



# Investigating the impacts of biochar on water fluxes in tropical agriculture using stable isotopes

Benjamin M. C. Fischer[1,2,3], Laura Morillas[4], Johanna Rojas Conejo[5], Ricardo Sánchez-Murillo[6],

Andrea Suárez Serrano[5], Jay Frentress[7, 8], Chih-Hsin Cheng[9], Monica Garcia[10], Stefano Manzoni[1,3],

Mark S. Johnson[11,12], and Steve W. Lyon[1,3,13]

[1]    Department of Physical Geography, Stockholm University, Stockholm, Sweden.

[2]    Department of Earth Sciences, Uppsala University, Uppsala, Sweden

[3]    Bolin Centre for Climate Research, Stockholm University, Stockholm, Sweden

[4]    Centre for Sustainable Food Systems, The University of British Columbia, Vancouver, British Columbia V6T 1Z4, Canada

[5]    Water Resources Center for Central America and the Caribbean (HIDROCEC-UNA), Universidad Nacional de Costa Rica, Guanacaste,

[6]    Stable Isotopes Research Group and Water Resources Management Laboratory, Universidad Nacional, Heredia, Costa Rica

[7]    Free University of Bolzano, Italy

[8]    Water Resources, Ramboll Sverige AB, Stockholm, Sweden

[9]    School of Forestry and Resource Conservation, National Taiwan University, Taipei, Taiwan

[10]    Department of Environmental Engineering, Technical University of Denmark, 2800 Kgs. Lyngby, Denmark

[11]    Department of Earth, Ocean and Atmospheric Sciences, The University of British Columbia, Vancouver, British Columbia V6T 1Z4 Canada

[12]    Institute for Resources, Environment and Sustainability, The University of British Columbia, Vancouver, British Columbia V6T 1Z4, Canada

[13]    School of Environment and Natural Resources, Ohio State University, Ohio, USA

*Correspondence to*: B. M. C. Fischer (benjamin.fischer@natgeo.su.se)

Keywords:        biochar, stable isotopes of water, soil and plant water, soil water retention curves, plant water uptake



## Abstract

Amending soils with biochar, a pyrolyzed organic material, is an emerging practice to potentially increase plant available water. However, it is not clear (1) to what extent biochar amendments increase soil water storage relative to non-amended soils and (2) whether plants grown in biochar amended soils access different pools of water compared to those grown in non-amended soils. To investigate these questions, we set up an upland rice field experiment in a tropical seasonally dry region in Costa Rica, with plots treated with two different biochar amendments and control plots, from where we collected hydrometric and isotopic data ($\delta^{18}O$ and $\delta^2H$ from rain, soil, groundwater and rice plants). Our results show that the soil water retention curves for biochar treated soils shifted, indicating that rice plants had 2 % to 7 % more water available throughout the growing season relative to the control plots. In addition, we observed a within treatment variability in the soil water retention curves which was in the same order of magnitude as one would expect from responses due to differences in biochar application rates or due to differences in biochar typologies. The stable water isotope composition of plant water showed that the rice plants across all plots preferentially utilized the more variable soil water from the top 20 cm of the soil instead of using the deeper and less variable sources of water. Our results indicated that rice plants in biochar amended soils could access larger stores of water more consistently and thus could withstand dry spells of seven extra days relative to rice grown in non-treated soils. Though supplemental irrigation was required to facilitate plant growth during extended dry periods. Therefore, biochar amendments can complement, but not necessarily replace, other water management strategies.



## 1. Introduction

Rainfed agriculture provides food for the growing world population (Fraiture et al., 2009; Fraiture and Wichelns, 2010) without over-exploiting groundwater resources (Famiglietti, 2014; Jasechko et al., 2017). However, the spatial and temporal variability of rainfall makes rainfed agriculture vulnerable to droughts (Fischer et al., 2013) and poses a risk for food security (Fraiture and Wichelns, 2010). Extreme weather events such as El Niño-Southern Oscillation (ENSO) influence global precipitation patterns and can bring prolonged dry spells that limit rainfed agriculture production. This is especially true in the tropics, where rainfall regimes are changing and will continue to change (Feng et al., 2013; Giorgi, 2006; Knutson et al., 2006), leading to more frequent long-term droughts (i.e. periods of more than 10 years with limited rainfall; Hidalgo et al., 2019). Climate projections for the Mesoamerican tropics suggest (1) decreases in rainfall during the wet season (May-November) of 10 % to 25 %; (2) expansion of the areas affected by mid-summer droughts; and (3) increases in temperature and extreme dry spells – all of which result in a net decrease of water availability (Imbach et al., 2018). Such a decrease in water availability could have significant impacts on rainfed agricultural production and food security globally. Therefore, to reduce societal exposure to risk, it becomes necessary to make rainfed agriculture more resilient to current and future climate variability.

Agricultural innovations can offer a pathway forward. Common innovations considered capturing rain (Biazin et al., 2012) or flood water (Castelli et al., 2018), plant and soil water conservation measures (Enfors and Gordon, 2007; Makurira et al., 2007; Vico and Brunsell, 2018) or introducing supplementary irrigation (Mutiro et al., 2006). Amending soils with biochar is an emerging practice in agriculture that could be useful for improving resilience to climate variability (Fischer et al., 2018). Biochar is a collective name for organic material (e.g. woody or herbaceous vegetation, crop residues or waste material) that is pyrolyzed in low-tech (Sundberg et al., 2020) or high-tech furnaces (Liu et al., 2016). The result is a charcoal with different material properties (e.g. particle size, pore structure, surface area and hydrophobicity) from the original feedstock. Biochar can be applied on the soil surface or incorporated in the soil where it alters the original soil matrix thereby changing the infiltration capacity (Blanco-Canqui, 2017; Lim and Spokas, 2018; Sun and Lu, 2014) and creating a multilayer



soil profile. The altered soil physical characteristics increase the soil water holding capacity and more
in general the amount of soil water stored at a given soil matric potential (Omondi et al., 2016).
However, despite documented positive effects of biochar amendments on agricultural productivity
(Kätterer et al., 2019; Novak et al., 2016), also negligible or no effects have also been observed (Fischer
et al., 2018; Jeffery et al., 2015, 2017; Nelissen et al., 2015; Reyes-Cabrera et al., 2017). These diverging
findings might be due to different biochar typologies (Fischer et al., 2018), but also to the fact that many
of the available studies are based on laboratory and pot experiments unable to mimic the variety of
processes occurring in agroecosystems at field scale (Agegnehu et al., 2017; Blanco-Canqui, 2017;
Zhang et al., 2016).
At the agroecosystem scale, soil water depends not only on the storage characteristics of the soil, but
also on variability of vertical fluxes resulting from rainfall and irrigation, evaporation, leakage and
runoff (Falkenmark, 1997; Rockström, 1999; Vico and Porporato, 2015). Thus, biochar impacts could
manifest themselves across the myriad pathways by which water can move through the soil-plant-
atmosphere continuum. Stable water isotopes can be a powerful tool to study how biochar additions
modify water stores and fluxes in agroecosystems. As part of the water molecule itself, the stable
isotopes of the water ($^{18}O$ and $^{2}H$) in combination with hydrometric data, are a proven tool to trace flow
pathways of water from rainfall (Fischer et al., 2017b) to evaporation (Benettin et al., 2018; Gonfiantini,
1986), through the (un)saturated zone (Jasechko et al., 2017; Koeniger et al., 2016; Sánchez-Murillo
and Birkel, 2016; Saxena, 1987), catchments (Fischer et al., 2017a; Klaus and McDonnell, 2013) and
more recently in the soil-plant-atmosphere continuum (Allen et al., 2019; Brooks et al., 2010; Dawson
and Ehleringer, 1991; McDonnell, 2014; Penna et al., 2018; Rothfuss and Javaux, 2017; Sprenger et
al., 2016).
Root water resembles the isotopic composition from the absorbed soil water from a specific location in
the soil profile (Berry et al., 2018), while, xylem water in the plant stem represents the isotopic
composition of all the soil profile within the root network (Dawson and Ehleringer, 1991; Penna et al.,
2018). To identify which water stores are available to vegetation, various potential water sources -e.g.,
rain (Fischer et al., 2019; Prechsl et al., 2014), soil water (Sprenger et al., 2015) and groundwater (Beyer



et al., 2016) are collected and analyzed for their stable isotope composition. The stable isotope
composition of the different collected water has allowed researchers to develop new theories whether
plants use soil-bound vs. mobile soil water pools (Brooks et al., 2010) or consume water from specific
soil layers that change over time (Berry et al., 2018; Beyer et al., 2016; Goldsmith et al., 2012; Koeniger
et al., 2016; Muñoz-Villers et al., 2020). Amin et al (2020) compared results from different stable
isotopes studies performed in natural catchments and deduced that plants in dry tropical climates
consume water from soil layers deeper than 50 cm. Beyond investigating natural ecosystems, stable
isotopes offer opportunities to study the sources of water in agroecosystems and quantifying the
efficiency of agricultural innovations.
Despite that stable isotopes have been used to a lesser extent in agricultural systems than in natural
systems to investigate plant water sources (Penna et al., 2020), there are successful studies done in
coffee (Muñoz-Villers et al., 2020), maize, wheat (Stumpp et al., 2009) and rice cultures
(Mahindawansha et al., 2018; Shen et al., 2015). In the case of rice, Shen et al. (2015) observed that
flooded rice consumed soil water from 0-15 cm deep, while Mahindawansha et al. (2018) found that
upland rice in dry conditions mostly consumed soil water from up to 50 cm deep except during the
maturing stage, when plants shifted to use water from the 10-30 cm soil depth. Based on this evidence,
we hypothesized that amending biochar into the top 10-30 cm of the soil, as it is commonly done, could
increase resilience to climate variability of upland rice in the tropics.
Our study seeks to test this hypothesis explicitly in a field experiment with upland rice in soil amended
with two different biochar types vs. a control treatment (no biochar) in a tropical seasonally dry region
in northwestern Costa Rica. We use a combination of hydrometric and isotopic data ($\delta^{18}$O and $\delta^{2}$H of
rain, soil, groundwater and rice plants) to target 1) to what extent do biochar amendments increase the
soil water storage relative to non-amended soils during the growing period of rice? and 2) do rice plants
grown in biochar amended soils access different pools of water compared to those grown in non-
amended soils?


## 2. Study site and experimental design

### 2.1 Study site

The biochar rice experiment was conducted at the Enrique Jímenez Núñez Experimental Station
(EEEJN) from the Instituto Nacional de Innovación y Transferencia en Tecnología Agropecuaria
(INTA) near the city of Cañas in the Guanacaste province of Costa Rica (Figure 1a). Soils at the
experimental site are loamy vertosols (Table A1) typically more than 2 m deep (Diogenes Cubero and
Maria José Elizondo, 2014). Guanacaste province is part of the Dry Corridor of Central America
(Sánchez-Murillo et al., 2020) and characterized by a seasonally dry tropical climate with marked dry
and wet seasons and limited temperature variability over a year (Birkel et al., 2017). The annual average
temperature at EEEJN-INTA is 27.4 °C. The dry season typically spans from mid-November to April
with virtually no rainfall. Wet season precipitation exhibits a bi-modal distribution dominated by the
influence of the Intertropical Convergence Zone with peaks occurring in May/June and
September/October. The moderate dry period between these two peaks is usually referred to as the mid-
summer drought (Magaña et al., 1999). The average annual rainfall in the area is approximately 1,547
$\pm$ 473 mm yr$^{-1}$ based on a 100-year observation record from a meteorological station ~10 km distance
of the experimental site (Figure 2a). The annual average actual evapotranspiration is around 1,100 mm
yr$^{-1}$ (Sánchez-Murillo and Birkel, 2016). In the last century, 70 % of the driest years in this region (i.e.,
years with less than 1,153 mm yr$^{-1}$ of rainfall, which is the 25$^{th}$ percentile of annual rainfall), occurred
during warm ENSO years. Based on the Standardized Precipitation Index (SPI; Naresh Kumar et al.,
2009), recurrent below average rainfall has been observed in this region since 1960s (Figure 2b) with a
significant periodicity of severe (SPI<-1.5) and sustained droughts of around 10 years (Hidalgo et al.,

147 2019).

### 2.2 Experimental design

For this experiment two types of biochar were tested to represent a more locally-produced biochar and
a more industrially-processed biochar, respectively. Biochar 1 (BC1) was made of locally sourced
bamboo (*Guadua angustifolia*) and produced at the Costa Rica Institute of Technology (TEC, Cartago,
CR; Table A1). The feedstock consisted of wood pieces up to 30 cm in length from construction waste,
which were pyrolyzed using a pyrolysis furnace under a temperature ranging 450-480 °C. A second



biochar, biochar 2 (BC2) was produced from sugarcane filter cake collected from the Huwei Sugar Mill
(Taiwan Sugar Corporation, Taipei, Taiwan). For the industrial processing of BC2, the filter cake was
pelletized into pellets with 7.6 mm diameter and 20-30 mm long and pyrolyzed at 600 °C under a
controlled nitrogen-rich atmosphere. Pyrolyzed pellets were crushed and sieved to $\leq 2$ mm prior to field
application.
Within the EEEJN-INTA experimental station, an area of approximately 160 $m^2$ was delineated and
divided into three sections of 40 $m^2$ each for treatments. The three different treatment sections, one for
each biochar type (BC1 and BC2) and a control treatment (C) with no biochar added, were subdivided
into three plots each to create three independent monitoring replicates of each treatment (Figure 1b).
The BC1 and C plots were 7 $m^2$ each (5 m long x 1.4 m wide) in area while the BC2 plots were 3.5 $m^2$
each (2.5 m long x 1.4 m wide) in area. This difference in areas between biochar treatments was due to
a lower amount of BC2 being available (shortage of feedstock at the biochar supplier) while securing a
similar application rate (1 kg $m^{-2}$) across biochar treatments. For the biochar treatments, the $\leq 2$ mm
particle size biochar was mechanically worked into the top 20 cm of the field prior to planting. It should
be noted that BC1 was incorporated into the field about six months earlier than BC2 due to logistical
constraints. BC1 addition was followed by an irrigated melon crop on the treated plot prior to our rice
experiment.
After the treatment sections were prepared, an upland rice variety Palmar 18 (*Oryza sativa* L.) was sown
simultaneously on the three sections on 18 July 2018 indicating the start of the experiment. For sowing,
5 cm deep longitudinal rills were created in all plots with a spacing of 25 cm. In each rill, rice seeds
were sown by hand of about 1 seed $cm^{-1}$, equivalent to 20 g $m^{-2}$. After sowing, the rills were covered
with soil. During the growing season, rice plants were primarily rainfed which is the standard procedure
for the predominant upland rice grown in the region. In some cases, where water sources for irrigation
are available, sporadic support irrigation is used by local farmers to support crops and avoid wither.
Due to prolonged dry spells that occurred during the study period, all experimental plots were irrigated
with 7 L $m^{-2}$ on July 22 and August 25 to assist germination and avoid plant drought damage
respectively on each date. Following typical regional crop management practices,  fertilizer (100 g $m^{-2}$



consisting of 10 % N, 30 % P, 10 % K in combination of 11 ml MEGAFOL® and 11 g magnesium
sulphate) and insecticide/herbicide (2 ml Muralla® Delta; 50 ml Garlon and 20 ml bispiribac sodium)
were applied to all experimental plots using 2 L m$^{-2}$ irrigation water on each treatment date (August 10,
September 6, and November 5) to support plant growth. At monthly intervals, manual weed control was
performed in all plots. Harvest took place on 21 November 2018 and indicated the end of the
experiment.
## 2.3 Instrumentation and sampling
### 2.3.1 Meteorological and hydrometric observations
A meteorological station (Vaisala WT520; 1.5 m height) was used to continuously monitor
precipitation, wind speed and direction, air temperature, relative humidity and atmospheric pressure at
the site during the entire study period (Figure 1b and c). Each experimental plot was instrumented with
one sensor installed at 15 cm depth to monitor volumetric soil water content, soil electrical conductivity
and soil temperature (model GS3, Decagon Devices, Inc., Pullman USA), and one additional sensor at
same depth to monitor soil matric potential and soil temperature (model MPS6, Decagon Devices). Both
sensors were between rice rows in each plot (Figure 1c). Additionally, soil samples were collected at
15 cm soil depth from each plot at the beginning of the experiment and after harvest to determine the
gravimetric soil moisture content. These data were used to perform a two-point calibration of the
volumetric soil water content measurements derived from the sensors at each plot during the entire time
series.
Depth of groundwater levels was measured using a groundwater well (groundwater well A) installed
between the BC1 and C treatment sections (Figure 1b). The well consisted of screened PVC tube
instrumented with a sensor to continuously monitor groundwater level, electrical conductivity and water
temperature (model CTD, Decagon Devices). Manual water level measurements were also made every
other week during the study period to calibrate the continuous sensor data. All sensors were connected
to a datalogger (Campbell CR1000 logger and an AM416 Relay Multiplexer) and programmed to record
at 30-minute intervals.



2.3.2 Water and plant sample collection
Water samples from different pools of water (namely, rainwater, irrigation water, soil water and
groundwater) were collected for isotopic analysis. Rainwater was collected using a funnel connected
with tubing to a PET bottle (1.5 liter) wrapped in aluminum foil similar to Prechsl et al. (2014). In each
plot, lysimeters (Soilmoisture equipment corp., Santa Barbara, USA) were installed in the soil reaching
to 15 cm and 40 cm soil depth respectively to sample soil water. Groundwater samples were collected
from a second groundwater well (groundwater well B) installed near the BC2 treatment section (Figure
1b).
Rainwater samples were collected daily at 7:00 AM. Water from additional application sources such as
irrigation (to supplement rainfall) and fertilizer/pesticide/herbicide applications were sampled as a grab
sample using a PE bottle during each application. Soil water and groundwater samples were collected
approximately biweekly (every other week) after plant germination from 31 July 2018 until the harvest
day on 21 November 2018, resulting in 11 sampling days. Soil water was collected from lysimeters by
applying an 800-mbar vacuum for 2 minutes. Groundwater was sampled by purging the well and
waiting 1 hour before collecting the groundwater sample. All water samples were collected in 30 ml PE
bottles, which were capped and sealed with Parafilm® for transport and cold storage (5 °C) until
analysis. At the end of each sampling day, all excess water from all sampler tubing, bottles, and suction
lysimeters was removed to prevent inter-sampling contamination.
Plant material from the rice plants was also collected on each of the 11 biweekly sampling dates at
around 12:00 noon. For plant material sampling, six rice plants were randomly selected within each
plot. The plant height from the soil to the plant tip was measured and recorded before sampling. To
avoid loss of biomass on sampled plants, the plants were extracted using a small knife which was
carefully wiggled into the soil. The roots, stems and leaves of the extracted plants were separated
immediately and transferred into double re-sealable zipper storage bag. To minimize post-sampling
transpiration, storage bags were directly placed in a cooler with ice. All plant material was stored in the
laboratory freezer (-80 °C) before extracting the plant water for isotopic analysis.





## 3. Laboratory methods and data analysis

### 3.1 Plant water extraction

Plant water was extracted from the stem (xylem water) of the different rice plants to infer which sources
of water the rice plants used. We used the cryogenic vacuum extraction technique described by
Koeniger et al., (2011) to extract the plant water for stable isotope analysis. The method uses a heated
vial and a cold trap vial (Exetainer® vial with standard cap and rubber septum, Labco Ltd, Lampeter,
United Kingdom) connected with stainless-steel capillary tubing. About 3 g of plant material from the
rice stem was placed in the heated vial before the system was evacuated to 85 kPa with a vacuum hand
pump (Mityvac). The heated vial was heated for 1 hour at 100°C using a test tube heater (HI839800
COD Test Tube Heater; Hanna instruments) while the cold trap vial rested in a Dewar flask containing
liquid nitrogen at about -196°C. After the extraction was stopped, the cold trap vial was sealed with
Parafilm and left to thaw. After thawing, the extracted liquid water was pipetted into 2 ml vials (32 x
11.6 mm screw neck vials with cap and PTFE/silicone/PTFE septa) and stored cold (5 °C) until stable
isotope analysis. On average 86±5 % plant water was extracted from xylem.

### 3.2 Isotope analysis

All non-plant water samples were filtered (0.45 µm filter 13 mm PTFE Syringe Filter, Fisher scientific)
and pipetted in vials (2 mL into a 1.5 mL 32 × 11.6 mm screw neck vials with cap and
PTFE/silicone/PTFE septa) prior to analysis. Water stable isotopes analysis was conducted at the Stable
Isotopes Research Group facilities of the Universidad Nacional of Costa Rica using a water isotope
analyzer LWIA-45P (Los Gatos Research Inc., USA). All data were normalized and corrected for drift
and memory effects. The analytical long-term error was ± 0.5 (‰) (1σ) for $δ^2H$ and ± 0.1 (‰) (1σ) for
$δ^{18}O$.
Plant water stable isotopes analysis was conducted at the Swedish University of Agricultural Sciences
(SLU) Stable Isotope Laboratory (SSIL) in Umeå using an Isotope Ratio Mass Spectrometer (TC/EA-
IRMS; DeltaV Advantage, Thermo Fisher Scientific, Bremen, Germany; High Temperature Conversion
Elemental Analyzer, Thermo Fisher Scientific, Bremen, Germany and an AI 1310 Autosampler,
Thermo Fisher Scientific, Bremen, Germany). All water samples were injected into a glassy carbon





reactor containing glassy carbon chips at 1,400°C and converted to $H_2$ and CO gases which were
separated on a column and analyzed on a mass spectrometer. All data were corrected for drift and
memory. The analytical precision and accuracy were ± 2 (‰) (1σ) for $\delta^2H$ and ± 0.15 (‰) (1σ) for
$\delta^{18}O$.
All stable isotope compositions are presented as delta notations (δ) in ‰, relating the ratios (R) of
$^{18}O/^{16}O$ and $^2H/^1H$, relative to the VSMOW-SLAP scale. The Global Meteoric Water Line (GMWL)
was defined as $\delta^2H = 8 \cdot \delta^{18}O + 10$ by Craig (1961). The Local Meteoric Water Line (LMWL) was derived
as $\delta^2H = 7.4 \cdot \delta^{18}O + 5.5$ using the long term isotopic data from the rain sampler at the Water Resources
Center for Central America and the Caribbean (Sánchez-Murillo et al., in review) located ~50 km
distance of the experimental site. In addition, the deuterium excess (d-excess) was defined as d-excess
$= \delta^2H - 8 \cdot \delta^{18}O$ (Dansgaard, 1964).
3.3 Evapotranspiration and soil water retention impacts
Daily evapotranspiration rates (ET) from the experimental area were estimated by the crop coefficient
method ( $ET = K_c \cdot ET_{ref}$ ) or FAO56 Penman-Monteith method (Allen et al., 1998). We used site specific
meteorological observations to estimate daily reference ET ($ET_{ref}$) and experimentally derived crop
coefficient ($K_c$) values for the three different stages of the crop growth (initial, mid-season, and late-
season). Instead of using globally averaged values of $K_c$ for rice (Allen et al., 1998), we used region-
specific $K_c$ values experimentally derived from a nearby field experimental site equipped with an Eddy
Covariance (EC) tower where the same variety of upland rice is grown (Morillas et al., 2019). Daily $K_c$
values from the EC site where derived as the ratio of daily measured ET and site-specific $ET_{ref}$, and then
averaged for the three stationary crop growth stages ($K_c$ initial = 0.7, $K_c$ mid-season = 0.9 and $K_c$ late
season = 0.5). The length of each crop growth stage was also calibrated for this region by observing the
pattern of daily measured ET over the whole growing season (initial ≈ 25 days, development ≈ 20 days,
mid-season ≈ 50 days, late-season ≈ 23 days for an average growing season of 120 days).
Field derived 30-minute records of all meteorological and hydrometric observations (precipitation,
volumetric soil water content, soil matric potential and groundwater level) were aggregated to daily
averages.  Accumulated  precipitation  and  evapotranspiration  were  also  derived  from  daily



measurements and estimates respectively for the entire experimental period (July 18-November 21).
Average volumetric soil water content and soil matric potential for each treatment (BC1, BC2 and C)
were calculated by averaging the observations in the three replicated plots per treatment.
Treatment specific volumetric soil water content ($\theta$) and soil matric potential ($\psi$) were linked through
soil water retention curves using the Van Genuchten model (Van Genuchten, 1980) (Eq. 1)

$$\theta = \theta_r + \frac{\theta_s - \theta_r}{[1 + (\alpha\,\psi)^n]^m} \qquad (1)$$

where $\theta_r$ [%], $\alpha$ [-] and $n$ [-] represent residual, and the fitted scale and shape parameters, respectively;
parameter and $m = 1 - 1/n$ [-] while saturation soil moisture ($\theta_s$) is based on field observations. To
examine the effect of biochar on soil physical and hydraulic properties, we compared the indicators $\theta_{WP}$;
$\theta_{FC}$ and van Genuchten parameter $\alpha$ and $n$ estimated for the biochar amended treatments (BC1 and
BC2) with the same indicators for the unamended treatment (C) using response ratios ($RR$) as in Fischer
et al. (2018). For this study, $RR$ represents the ratio of the variable of interest in the treatment to the
same property in the control such that $RR>1$ or R<1 indicates that the treatment has a positive or
respectively negative effect.

## 3.4 Plant water source estimation

The isotopic composition of the water samples was represented in the dual isotope space $\delta^{18}O$ and $\delta^2H$
to infer which sources of water rice plants consumed. To represent a potential plant water source under
rainfed conditions, the isotope composition of rainfall was considered as the volume weighted isotope
composition of rainfall collected in the two-week period before a given plant water sampling day. Since
residual rainfall can evaporate while in the soil (simplified assumption not accounting of mixing with
pre-event water), the isotopic composition of the residual rainfall for each water sampling day was
estimated following (Gonfiantini, 1986) and (Benettin et al., 2018)

$$\delta_{PR} = (\delta_P - \delta^*)(1 - f_E)^U + \delta^* \qquad (2)$$

where $\delta_{PR}$ [‰], $\delta_P$ [‰], and $f_E$ [-] represent the isotopic compositions of the residual rainfall, the
volume weighted isotope composition of rainfall collected in the two-week period before a sampling





day, and the fraction of rainfall that fell in the two-week period before a sampling day and that has
evaporated on the sampling day, respectively. The variables $\delta^*$ [‰] and $U$ [-] represents the limiting
isotopic composition and the temporal enrichment slope, which were determined using equation 3 and
4 respectively

$$\delta^* = \frac{R_H \delta_A + \varepsilon_k + \frac{\varepsilon^+}{\alpha^+}}{R_H - 10^{-3}\left(\varepsilon_k + \frac{\varepsilon^+}{\alpha^+}\right)} \qquad (3)$$

$$U = \frac{R_H - 10^{-3}\left(\varepsilon_k + \frac{\varepsilon^+}{\alpha^+}\right)}{1 - R_H \varepsilon_k} \qquad (4)$$

where $R_H$ [-] represents the average relative humidity of the two-week period before a sampling day, $\delta_A$
[‰] the approximation of the isotopic composition of the atmospheric vapor (equation 5 following
Gibson et al., (2016)), $\varepsilon_k$ [‰] the simplified kinetic fractionation factor (Eq. 6) and $\varepsilon^+$ [‰] and $\alpha^+$ [-] the
two equilibrium fractionation factors (Eq. 7 and 8).

$$\delta_A = \frac{\delta_P - \varepsilon^+}{\alpha^+} \qquad (5)$$

$$\varepsilon_k = (1 - R_H)(1 - S_{18O\ or\ 2H})10^3 \qquad (6)$$

$$10^3 ln\left(\alpha^+ \delta\ ^2H\right)$$
$$= 1158.8\frac{T^3}{10^9} - 1620.1\frac{T^2}{10^6} + 794.84\frac{T}{10^3} - 161.04$$
$$+ 2.9992\frac{10^9}{T^3}$$

or                                                                                                     (7)

$$10^3 ln\left(\alpha^+ \delta^{18}O\right) = 0.3504\frac{10^9}{T^3} - 1.6664\frac{10^6}{T^2} + 6.7123\frac{10^3}{T} - 7.685$$

$$\varepsilon^+ = (\alpha^+ - 1)10^3 \qquad (8)$$



where $S_{18O} = 0.9755$ and $S_{2H} = 0.9723$ (Merlivat, 1978) and $T$ [K] represents the average
temperature of the two-week period before a sampling day. The volume-weighted isotope composition
of rainfall before each sampling day, which was generally near the GMWL and LMWL, and the
corresponding estimated isotopic composition of the residual rainfall, which was generally off the
GMWL and LMWL, provided the start and end point of a theoretical evaporation line in dual isotope
space. Similarly, an evaporation line for the median sampled soil water of a period was developed. Such
evaporation lines map the evolution of the soil water available from residual rainfall or evaporated soil
water for plants to be consumed between sampling days allowing us to track which stores of water the
rice plants interact with across the treatments. In addition, the within treatment variability defined as
difference between the minimum and maximum observed isotopic composition of plant water within a
treatment on any given sampling day were calculated.

## 4. Results

### 4.1 Hydrometric variability

Based on the temporal variability of rainfall, we identified three distinct periods within the overall study
period (Figure 3). Period I (18 July to 20 September) was characterized with alternating wet and dry
days, Period II (20 September to 9 November) presented consistent high daily rainfall inputs, and Period
III (10 November to 21 November) was characterized by a long dry spell ending with rice harvest.
Throughout the study period, daytime air temperatures were around 26.7 °C (standard deviation = 3 °C)
and evapotranspiration rates on average 3.1 mm day$^{-1}$ (standard deviation = 0.7 mm day$^{-1}$).
During Period I (germination and vegetative phase), the rice in the different plots grew to a height of
50 cm in all experimental plots (standard deviation <2.5 cm). This period was characterized by
intermittent dry and wet spells with accumulated precipitation slightly higher than evapotranspiration
($P_{cum}$= 240 mm and $ET_{cum}$= 191 mm over the 64-day period; Figure 3b). During this period, the
maximum recorded volumetric soil water contents were 40 %, 43 %, and 35 %, and decreased to the
minimum values 30 %, 25 %, and 23 % in the BC1, BC2, and control treatment, respectively (Figure
3c). Regarding soil matric potential ($\psi$) during this period, it surpassed field capacity ($\psi_{FC}$ = -0.05
MPa) with a maximum of -0.008 MPa during rain events and decreased to a minimum of -0.32 MPa



observed in all treatments a few days after the third sampling day as a result of the driest spell of Period
I (Figure 3d). Generally, the soil matric potential in the biochar treatments was 0.002 MPa higher than
in the control treatment and never reached the wilting point ($\psi_{WP}$ = -1.5 MPa). The groundwater level
was generally 0.7 m below the surface, rising after sampling day 1 to less than 0.6 m below the surface
before sampling day 4, and to less than 0.5 m below the surface in response to the largest rainfall of
Period I (Figure 3e).
During Period II (vegetative and reproductive phase), rice plants attained their maximum heights of
around 100 cm (standard deviation <5 cm), across all three plots (Period II was the wettest period with
15 out of 42 rain days with intensities greater than 20 mm d$^{-1}$ of rainfall (and one day with 93 mm d$^{-1}$)
(Figure 3a and b). This wet condition lead to cumulative precipitation being much greater than
cumulative evapotranspiration during the period ($P_{cum}$= 570 mm and $ET_{cum}$= 147 mm; over the 50-day
period). The volumetric soil water content over Period II was generally higher than in Period I, with
multiple peaks driven by rainfall events and then a decrease towards the end of the period. After rain
events, the volumetric soil water contents rose from 28 % to 40 %, from 24 % to 45 %, and from 23 %
to 36 % in BC1, BC2, and control treatment, respectively. Soil moisture then decreased in the three
treatments to 32 %, 38 %, and 32 % during the last part of the period. The soil matric potential during
Period II remained largely above field capacity except by the end of the period when it decreased (before
sampling day 8) to a minimum of -0.23 MPa in BC1 and -0.16 MPa in BC2 and C. The groundwater
level increased multiple times during this period from 0.7 m below the surface to reach the soil surface
the rainiest day of the study period. Between Sampling days 6 and 7, groundwater level remained no
lower than 0.4 m below the surface.
During the final experimental period, Period III (ripening phase), rice plants maintained their maximum
height acquired by the end of Period II. This period was characterized by a 12 day long dry spell such
that cumulative evapotranspiration was greater than cumulative precipitation ($P_{cum}$= 2 mm and $ET_{cum}$=
63 mm; 12-day period). By the end of Period III, the volumetric soil water content in the BC1 and BC2
treatments converged to the lowest observed value of ~21 %. It is relevant that the control treatment
reached this value about seven days earlier than the biochar amended plots, and the control plots



continued decreasing to reach a minimum value of 18 % (Figure 3c). The soil matric potential for all
three plots decreased from above the field capacity to near the wilting point by the end of Period III.
The groundwater level also decreased from 0.4 m to 0.8 m below the surface (i.e. the sampling well
went dry).
## 4.2 Impact of biochar on soil water retention curves
The soil water retention curves from the different treatments showed different shapes and different
volumetric water content at a given soil matric potential (Figure 4). Comparing the different soil water
retention curves across the different plots of the different treatments shows a within treatment
variability, i.e., range of different volumetric soil moisture contents relative to the observed soil matric
potentials (Figure 4). Comparing the different soil water retention curves across the periods shows that
biochar treatments increased volumetric soil moisture content relative to the control treatment
consistently across the ranges of observed soil matric potentials in all three periods (Figure 4, Table
A2). The soil water retention curves estimated for Period III were shifted to lower volumetric water
contents relative to the other periods and ranged from close to field capacity to wilting point.
The effect of biochar on the soil water retention curve can also be quantified by the response ratios of
the wilting point, field capacity and the van Genuchten parameters $\alpha$ and n. Most of these ratios were
found to be larger than one (Table 1), which indicates increased soil water content for a given water
potential value.
## 4.3 Isotopic variability
Overall, the $\delta^{18}O$ and $d$-excess of rainfall was between -15.7 ‰ and -0.2 ‰ ($S_D$ = 3.4 ‰) and 0 ‰ and
+18 ‰ ($S_D$ = 4.6 ‰) respectively ($\delta^{18}O$ see Figure 5a, $d$-excess see Figure A2a and A3). The $\delta^{18}O$ and
$d$-excess of soil water and groundwater collected on the different sampling days was between -7.5 ‰
and -4.5 ‰ ($S_D$ = 1.3 ‰) and -1.1 ‰ and +9.7 ‰ ($S_D$ = 4.9 ‰) respectively ($\delta^{18}O$ see Figure 5b-d, $d$-
excess see Figure A2b-d and A3). The within treatment variability in isotopic composition of soil water
samples for each sample day was <1 ‰ for $\delta^{18}O$ and <6 ‰ for $d$-excess (Figure 6). The $\delta^{18}O$ and $d$-
excess of plant water was between -8.7 ‰ and -2.7 ‰ ($S_D$ = 3.7 ‰) and -14.6 ‰ to +3.2 ‰ ($S_D$ = 11.4
‰) respectively ($\delta^{18}O$ see Figure 5b-d, $d$-excess see Figure A2b-d and A3). The within treatment

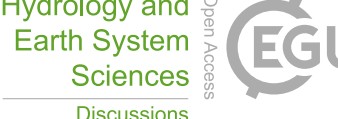

variability in isotopic composition of plant water samples on each sample day >3 ‰ for $\delta^{18}$O and >8 ‰
for *d*-excess (Figure 6). The within treatment variability was smaller for the biochar amended treatments
relative to the within treatment variability in the control treatment (Figure 6).
During Period I, the isotopic composition of rainfall varied between -5.6 ‰ to -0.2 ‰ for $\delta^{18}$O (Figure
5a) and from -1.1 ‰ to +9 ‰ for *d*-excess (Figure A2). On rainy days when rainfall intensities were
below 10 mm d$^{-1}$, sub-cloud evaporation may exert an important control on rainfall enrichment
(Sánchez-Murillo et al., 2016, 2017) and potentially also the low amount of rain water collected in
relation to the bottle volume causing water to evaporated water in the sampler. For example, the
observed fractionated isotopic compositions of these rain samples were often recorded to be <5 ‰ with
regard to *d*-excess. The average isotopic composition of plant water in the different treatments decreased
from roughly from +3.2 ‰ to -4 ‰ for $\delta^{18}$O and increased from roughly -40 ‰ to +18 ‰ for *d*-excess
during Period I (Figures 5 and A2). In Period II, the isotopic composition of rainfall varied between -
3.7 ‰ to -12.7 ‰ for $\delta^{18}$O (Figure 5a) and +6 ‰ to +11.8 ‰ for *d*-excess (Figure A2). The average
isotopic composition of plant water varied in all treatments to between -7 ‰ to -2 ‰ for $\delta^{18}$O and -11.8
‰ to +9.2 ‰ for *d*-excess. It should be noted that there was a change from negative to positive *d*-excess
for the plant water isotopic compositions between sampling day five and seven, indicating a change
from highly fractionated isotopic compositions to compositions similar to that of rainfall. During the
dry spell of Period III no rainfall occurred and hence no rainwater was collected. Also, no soil water
could be extracted from lysimeters sampling water from 15 below the surface on sampling day 10 and
day 11. The average isotopic composition of plant water varied between -7 ‰ to -6 ‰ for $\delta^{18}$O and -7
‰ to -2 ‰ for *d*-excess, showing a high fractionation signature (Figures 5, A2 and A3).
### 4.4 Using dual isotope space to characterize plant water sources
Rainfall isotopic compositions fell along the GMWL and LMWL for our experimental site (Figure 7).
The soil water and ground water isotopic samples from Period I were more fractionated, i.e. they
deviated from the GMWL, compared to soil water isotopic samples from the wet Period II and III which
fell more along the GMWL (Figure 8). The plant water isotopic compositions from the different
treatments and sampling periods were somewhat different from each other in terms of absolute values
but showed a similar temporal evolution (Figure 7). In Period I, plant water samples from all treatments
deviated from the GMWL and moved primarily along the modeled evaporation lines of the sampled
soil water (Figure 7 a, d and g). The plant water thus resembled soil water with a strong evaporation
signature in Period I.
In Period II, which was much wetter than Period I, the plant water samples fell on or were close to the
GMWL independent of the treatment and moved from sampling day to sampling day along the GMWL.
It is likely that plant water responded to the replenished soil water that acquired the signature of rainfall
during this period. At the end of Period II, plant water samples from BC1 and the control treatment
showed a more fractioned signature and fell on the modelled evaporation line indicating that plant water
resembled soil water with signature from evaporated rain from day 8 (Figure 7 b and h). Plant water
samples in the BC2 treatment, however, showed the signature from soil water more similar to original
rainfall (Figure 7 e and e2). During the dry Period III, all plant water samples deviated from the GMWL
and fell along modeled evaporation lines with signature of residual rainfall that had fallen in Period II
(depicted in blue in Figure 7 c, f, and i).

## 5. Discussion

### 5.1 Variable effect of biochar on the soil hydraulic properties

Incorporating two different types of biochar in plots planted with rice affected the soil hydraulic
properties. The soil water retention curves of the biochar amended treatments showed higher soil water
contents at similar matric potential relative to the control treatment, leading to more plant water
available under similar conditions (Figure 4, Table 1). The soil water retention curve of the BC1
treatment became more similar to the curves found in finer grained soils, which indicates increased
water retention, a common expected impact of biochar additions (Fischer et al., 2018; Sun and Lu,
2014). Conversely, the soil water retention curve for the BC2 treatment became more similar to the
curves associated with coarser soils indicating enhanced water flows, which has also been described as
a potential impact of biochar additions (Fischer et al., 2018; Liu et al., 2017).
The overall soil response to biochar amendments in our experiment had a within treatment variability
but was comparable to the response found in other tropical soils where a lower range of $\theta_{WP}$ and $\theta_{FC}$



was found (Obia et al., 2016). But the soil water retention curves we found were more irregular shaped
compared to laboratory derived soil water retention curves reported in the literature (Iiyama, 2016;
Morgan et al., 2001) which usually present one single continuous drying curve (e.g. Batool et al., 2015;
Gląb et al., 2016 or Obia et al., 2016). Instead the field-data derived soil water retention curves in the
present study were field derived and the result of temporally variable atmospheric forcing. Specifically,
our observed within treatment variability in the soil water retention curves was a same order of
magnitude as the responses due to differences in biochar application rates or due to differences in
biochar typologies reported in laboratory studies (e.g. Batool et al., 2015; Gląb et al., 2016 or Obia et
al., 2016). Laboratory studies may overestimate the volumetric soil moisture content at a given soil
matric potential compared to field-derived soil water retention curves (Iiyama, 2016; Morgan et al.,

463    2001).

Although the two biochar types tested were produced in different ways, their experimental application
was similar (i.e. same application rate, similar particle size, application amount, depth, site
characteristics and climate). One key distinction between the two biochar treatments was the application
date, which may be important because aging can change the physical and chemical characteristics of
biochar (Blanco-Canqui, 2017). Due to some logistical constraints, biochar was introduced to the BC1
plot about six months before the BC2 plot. This allowed the biochar to age in situ and for the disturbed
soils to settle under the BC1 treatment. Thus, the BC2 soil likely had relatively larger macropores that
could have increased the connectivity of the 20 cm soil layer where biochar was applied with deeper
soil layers. This difference in application timing may have influenced the hydraulic differences in
results observed between the two biochar treatments (Figure 4) and amplified the differences due to the
contrasting production methods. Clearly, the interplay of all the possible biochar variables with all the
possible site-specific heterogeneities makes it challenging to isolate the biochar effect in
agroecosystems. Taken altogether, these differences in biochar treatment responses and the relative
impacts of both B1 and B2 biochar treatments compared to the control plot highlights the potential for
variability in biochar responses – which has been documented in the literature (Fischer et al., 2018) and
creates ambiguity around predicting the response of biochar amendments at field scale. This further



highlighting the difficulty to transfer laboratory-scale results to the field scale where management
decisions are made.
## 5.2 Temporally variable soil water fluxes
The isotopic composition of different water samples was useful to infer how water fluxes varied through
time. The isotopic composition of soil water sampled at two different depths across the plots was rather
stable over time compared to the temporally variable isotopic composition of rainfall (Figure 5). In
addition, the temporal variability of isotopic composition of soil water from our experiment was less
than the spatial variability or change in isotopic composition with depth reported in previous biochar
studies (e.g. Beyer et al., 2016; Koeniger et al., 2016; Saxena, 1987 and Sprenger et al., 2016). When
comparing our findings with other tropical systems, the $d$-excess of the soil water we found during dry
spells (Figure A2) had a smaller variation range than observed in a coffee plantation in Mexico by
Muñoz-Villers et al. (2020) and was generally less variable than observations made by Jiménez-
Rodríguez et al. (2020) in a tropical wet forest in Costa Rica. The low $d$-excess values and ranges of
the soil water observed in this study indicate high evaporative processes in the top soil layer (Amin et
al., 2020; Sprenger et al., 2016). This is consistent with our high estimated evapotranspiration rates
(average 3.1 mm day$^{-1}$ up to 6 mm day$^{-1}$) which are typical for the Dry Corridor of Central America
characterized by high solar radiation and air temperatures (Morillas et al., 2019).
During Period I, when rice plants were small and sparse, leaving much bare soil, the evaporation
occurring from the soil across the different treatments was homogenous, creating a low $d$-excess signal
in the soil water. During wet spells in Period II, the $d$-excess increased slightly, indicating mixing of
rainfall with soil water. At the end of Period II and throughout Period III, the $d$-excess remained higher
despite high evaporation, which might be due to a more homogenous crop cover creating a consistent
microclimate as described by Sprenger et al. (2017). The isotopic composition of groundwater (1) had
$d$-excess values similar to that of meteoric water during dry spells and (2) decreased during wet spells
showing a high evaporative signal (Figure A2). Such observed changes in $d$-excess are generally not
found in temperate zones (Sprenger et al., 2016), but indicate that rainfall flushed the fractionated soil
water downwards promoting mixing with groundwater (Gat and Airey (2006).


### 5.3 Temporally variable plant water sources

The studied rice plants had different water sources available during different periods of the experiment, but what water did they consume?

It is likely that the fractionation observed in the plant water collected in this study represents fractioned soil water that was consumed by the plants. This is consistent with results observed in previous studies using stable water isotopes to map out plant water sources (Brooks et al., 2010; Penna et al., 2020; Sprenger et al., 2016). Further, this interpretation of plant water composition is supported by plant water samples falling along the theoretical evaporation lines estimating how soil water would evolves isotopically due to evaporation. Therefore, it is likely that during Period I, the young rice plants (with shallow root system <20 cm as reported by Mahindawansha et al. 2018) consumed the fractionated soil water (Figure 7) which was not sampled with the lysimeters at 15 cm and 40 cm below the surface.

During Period II, plants grew to their maximum heights with roots reaching deeper soil layers (length >60 cm as reported by Mahindawansha et al. 2018). This means that the rice plants, similar to larger vegetation e.g. trees (Allen et al., 2019), would have had access to deeper and more-stable pools of water with a distinct lower $d$-excess signature. However, the isotopic composition of plant water during this period followed the GMWL (Figure 7 b, e and h), indicating that plants consumed largely shallow soil water from recent rainfall. In Period III, it became increasingly difficult to extract water from lysimeters at 15 cm below the surface and the isotopic composition of plant water drifted from the GMWL, along the theoretical evaporation line of residual rainfall which fell in Period II. With the experiment being held in the tropics and based on Amin et al (2020) one would expect that the rice plants with their longer roots would accessed access the more stable and older water stores in deeper subsurface zones below 60 cm. Instead, the rice plants in the different treatments preferably consumed the temporally variable and isotopically labeled newer surface soil water similarly to what has been documented in natural ecosystems (e.g. van der Velde et al., 2015) and temperate grasslands (Bachmann et al., 2015).

By mixing biochar in the top soil, a multi-layer soil profile was created and based on studies in natural catchments, e.g. Penna et al. (2018) or Sprenger et al. (2016), these different layers could store not only


different quantities of water but also water characterized by different ages. Performing additional
isotopic experiments (Beyer et al., 2016), higher temporal resolution sampling of plant water (Marshall
et al., 2020; Volkmann et al., 2016) and spatiotemporal soil water (Sprenger et al., 2015) or including
interception, transpiration and atmospheric processes into the experimental analysis (Jiménez-
Rodríguez et al., 2020) would allow to not only distinguish in more detail whether the rice plants prefer
bounded or mobile water (Berry et al., 2018; Brooks et al., 2010; McDonnell, 2014) but also to quantify
the fraction of water sources (Muñoz-Villers et al., 2020). Consequently, this would also allow to
indicate how long the soil water resides in the different soil layers before it is consumed by plants. In
addition to the aforementioned vertical processes also the lateral water fluxes (Sprenger and Allen,
2020) need to be considered to assess the field-scale responses to biochar amendments (Fischer et al.,
2018). These analyses are beyond the scope of this initial investigation; however, our results indicate
that rice plants growing in biochar amended soils not only had access to more water (Figure 4) but also
had a more stable source of green water (i.e. soil moisture from rainfall) and thus could withstand dry
spells seven days longer (Figure 3). Regardless of the potential advantages, as stated by Fischer et al.
(2018), it must be noted that biochar as water management tool does not adhere to a one size fits all
approach but needs fine tuning in accordance with climate, site and plant characteristics to obtain stable
and optimal yields.

## 6. Conclusions

Amending soils with biochar is an emerging and promising practice to improving resilience of rainfed
agriculture to climate variability by increasing the soil water and plant available water. Using an
experimental field study, we observed biochar amendments to create generally 2 % to 7 % higher soil
water content and therefore more plant water relative to the control treatment, despite differing impacts
between biochar treatments depending on the type of biochar and timing of application. In addition, we
observed a within treatment variability in the soil water retention curves which was in the same order
of magnitude as one would expect from responses due to differences in biochar application rates or due
to differences in biochar typologies. Further, we were able to trace the effect of biochar on the soil water
storage to investigate which water plants consume. The isotopic composition of soil water sampled in



two distinct depths in the different plots was rather stable in time compared to the temporal variable
isotopic composition of rainfall. The stable isotope composition of plant water instead showed that the
rice plants preferably consumed the temporal variable soil water comprised of residual rainfall the
experienced evaporation in the top 20 cm of the soil. When comparing the different treatments, our
results indicated that rice plants grown in biochar amended soils not only had more water available but
also had a more stable source of green water. Thus, these rice plants in biochar amended soils could
withstand dry spells of up to an extra seven days. Despite these positive effects of biochar amendment,
it still seems necessary to provide additional irrigation to facilitate optimal plant growth if extended dry
periods occur during certain growing stages to have optimal yields. So, while our study highlights some
of the usefulness of combining hydrometric and isotopic data to map out how biochar additions impact
plant-water interactions in the field, we acknowledge more work is needed to fully characterize the
influence biochar additions may have at scale on agroecosystems. This further understanding is
important given the need of more specific management recommendations to ensure biochar additions
in agricultural landscapes result in net benefits for both farmers and the environment.

## Data availability

Upon acceptance, all of the research data that were required to create the plots will be available from
the Bolin Center for Climate Research.

## Author contribution

BF, LM, MG, SM, MJ, AS and SL designed the experiment, and BF, JR carried it out. CC provided
BC2, RS analyzed the stable isotope composition of the collected water. BF performed the data analysis
and prepared the paper with contributions from all co-authors.

## Competing interests

The authors declare that they have no conflict of interest.

## Acknowledgements

We thank all the people who helped in the field and the laboratory, particularly Sharon Arce, Johnny
Arriola and Eduardo Rodríguez and all the HIDROCEC team of Universidad Nacional, Liberia, Costa



Rica. The authors would also like to thank the collaboration from the Stable Isotopes Research Group
& Water Resources Management Laboratory (Universidad Nacional, Heredia) on helping with the
lysimeters and wells installation as well as water stable isotopes. Especially Edwin Quirós Ramos,
Roberto Ramírez, Juan Carlos Jiménez Vargas and all technical staff from the EEEJN-INTA who help
develop the experimental design and advised about regional crop management practices and Dr. Jaime
Quesada from TEC for providing the biochar national used in this study.

## 593 Financial support

This research was conducted as part of the Agricultural Water Innovations in the Tropics (AgWIT)
project funded by the Joint Call of the Water Joint Programming Initiative (Water JPI) and the Joint
Programming Initiative on Agriculture, Food Security and Climate Change (FACCE-JPI) of the
European Union and partner countries. Stefano Manzoni and Steve Lyon acknowledge partial support
from the Swedish Research Agencies Vetenskapsrådet, Formas, and Sida through the joint call on
Sustainability and resilience-Tackling climate and environmental changes (grant VR 2016-06313), and
the Bolin Centre for Climate Research (Research Area 7). Ricardo Sánchez-Murillo acknowledges the
financial support from the International Atomic Energy Agency (IAEA) grants COS/7/005, RC-19747
(CRP-F31004), RC-22760 (CRP-F33024) which were fundamental to conduct the water stable isotope
analysis in Costa Rica.



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



# Table
***Table 1***     *Response ratios for wilting point ($\theta_{WP}$), minimum observed average volumetric soil moisture contents ($\theta_{min}$) field*
*capacity ($\theta_{FC}$), and for the van Genuchten parameters $\alpha$ and n (Equation 1) for BC1 and BC2. Parameters are*
*derived for the average soil water retention curve of figure 4 for Periods I-III. A response ratio RR > 1 indicates*
*that biochar has a positive effect on a soil water content while a RR ≈ 1 indicates that biochar has no effect,*
*while RR < 1 indicates a negative response for the variable of interest.*

|  | **BC** | **Period I** | **Period II** | **Period III** |
|---|---|---|---|---|
| $\theta_{WP\ BC}\ \theta_{WP\ C}^{-1}$ | 1 | 1.36 | 1.46 | 1.18 |
|  | 2 | 1.16 | 1.32 | 1.03 |
| $\theta_{min\ BC}\ \theta_{min\ C}^{-1}$ | 1 | 1.12 | 1.16 | 1.17 |
|  | 2 | 1.08 | 1.03 | 1.11 |
| $\theta_{FC\ BC}\ \theta_{FC\ C}^{-1}$ | 1 | 1.08 | 1.14 | 1.04 |
|  | 2 | 1.13 | 1.13 | 0.88 |
| $\alpha_{BC}\ \alpha_{C}^{-1}$ | 1 | 1.29 | 0.50 | 0.68 |
|  | 2 | 3.21 | 1.34 | 1.08 |
| $n_{BC}\ n_{C}^{-1}$ | 1 | 1.00 | 0.89 | 1.47 |
|  | 2 | 1.00 | 0.92 | 1.06 |


Figures

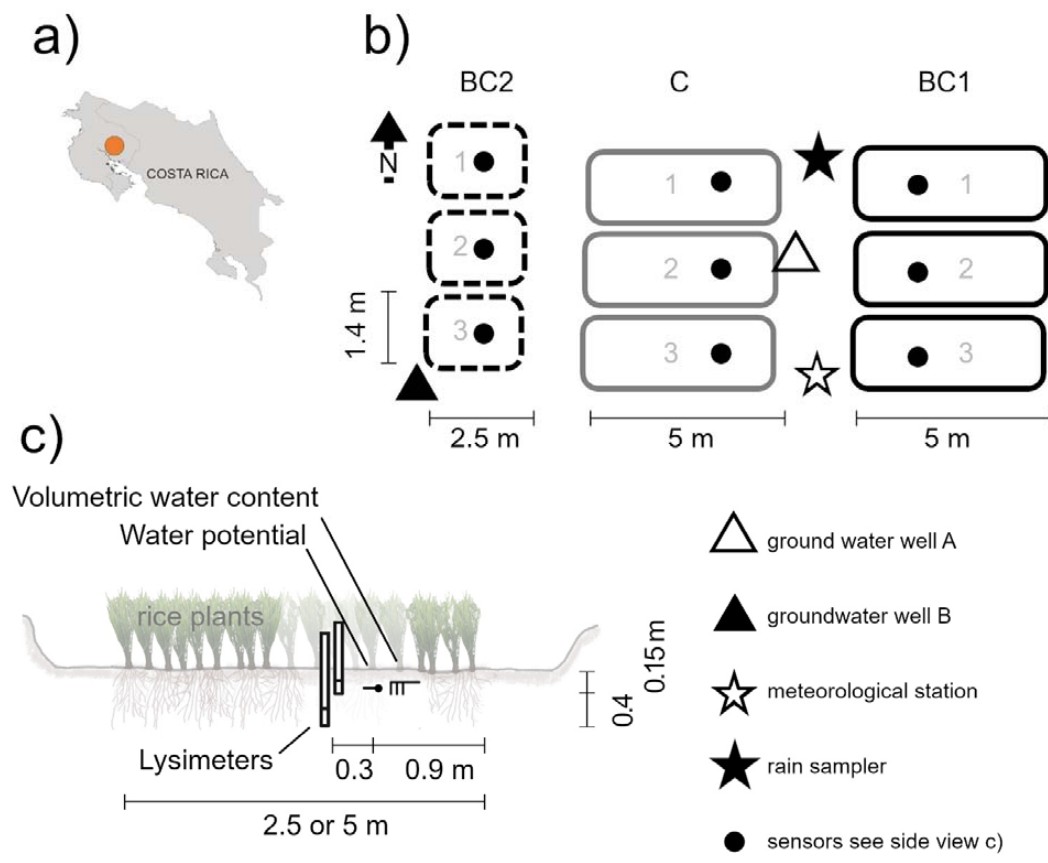


**Figure 1**  **(a) Map of Costa Rica with location of the experimental site (orange circle), (b) schematic top view of the rice**
**experiment with the three different treatment sections, BC1, BC2 and C. Symbols indicate the different**
**instruments: rain sampler for stable isotope samples (filled star), meteorological station (open star), continues**
**groundwater level measurements in well A (open triangle), groundwater well B for stable isotope samples**
**(closed triangle) and (c) a schematic side view of a plot with suction lysimeters for stable isotope samples 15 cm**
**and 40 cm below the surface, the water potential and volumetric water content sensors.**





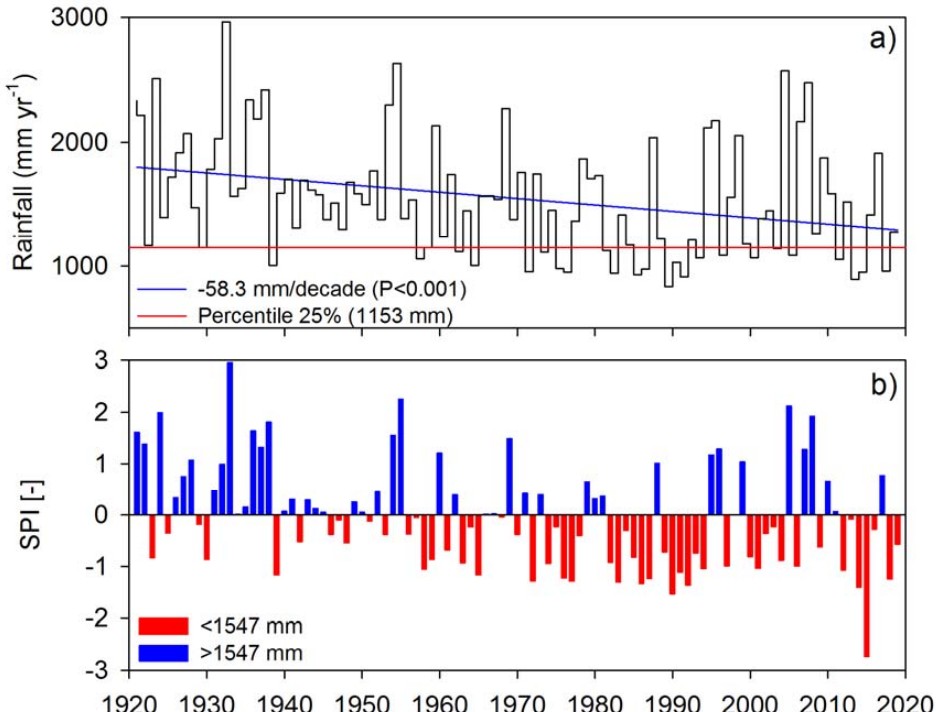


**Figure 2** *(a) Long-term rainfall (mm yr⁻¹) including a significant rainfall decrease of -53 mm per decade (blue line) and*
*25% percentile of 1153 mm (red line as reference) and b) Standardized Precipitation Index (SPI) within the*
*lowlands of Guanacaste between 1921-2019 (Long-term rainfall average=1547±473 mm yr⁻¹)(Rainfall data*
*source: Ing. Werner Hagnauer, Cañas, Guanacaste).*





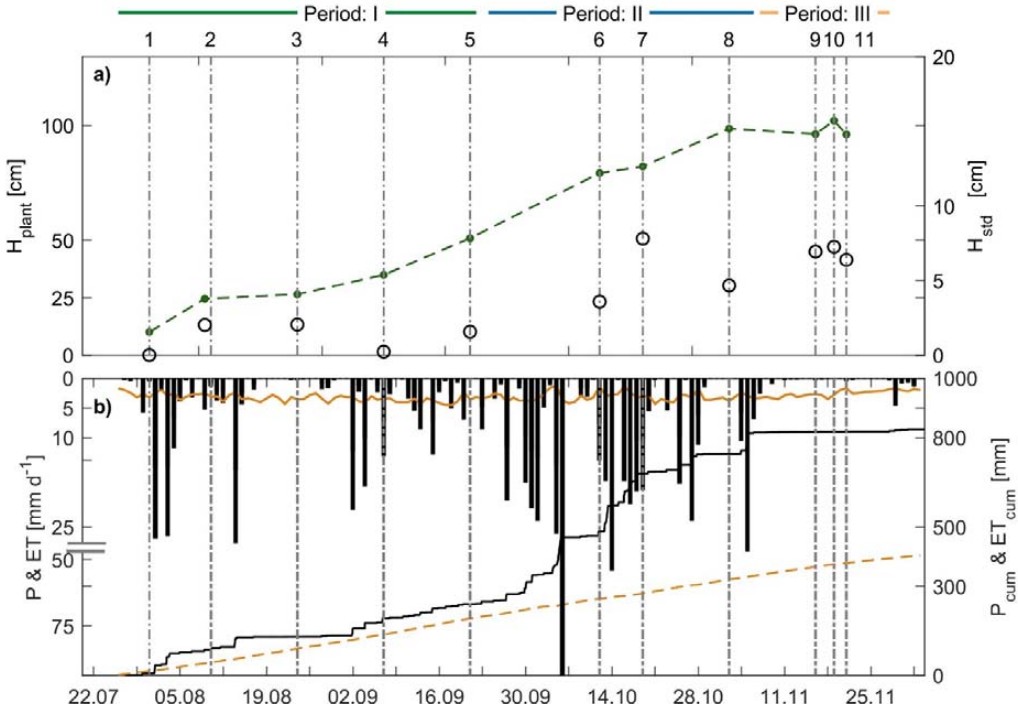


**Figure 3** *Time series of (a) rice plant average height ($H_{plant}$) of the rice plants (filled green circles and dashed line) and the standard deviation the plant height (open black circles); b) precipitation (P, black bars), estimated evapotranspiration (ET, solid orange line), accumulated P (solid black line) and accumulated ET (orange dashed line). The different water sampling days 1-11 are indicated in each panel as vertical dashed lines and numbered on top of panel a and the date are given on the x-axis of panel b as dd.mm. Period I, II and III are indicated on the top of panel c.*



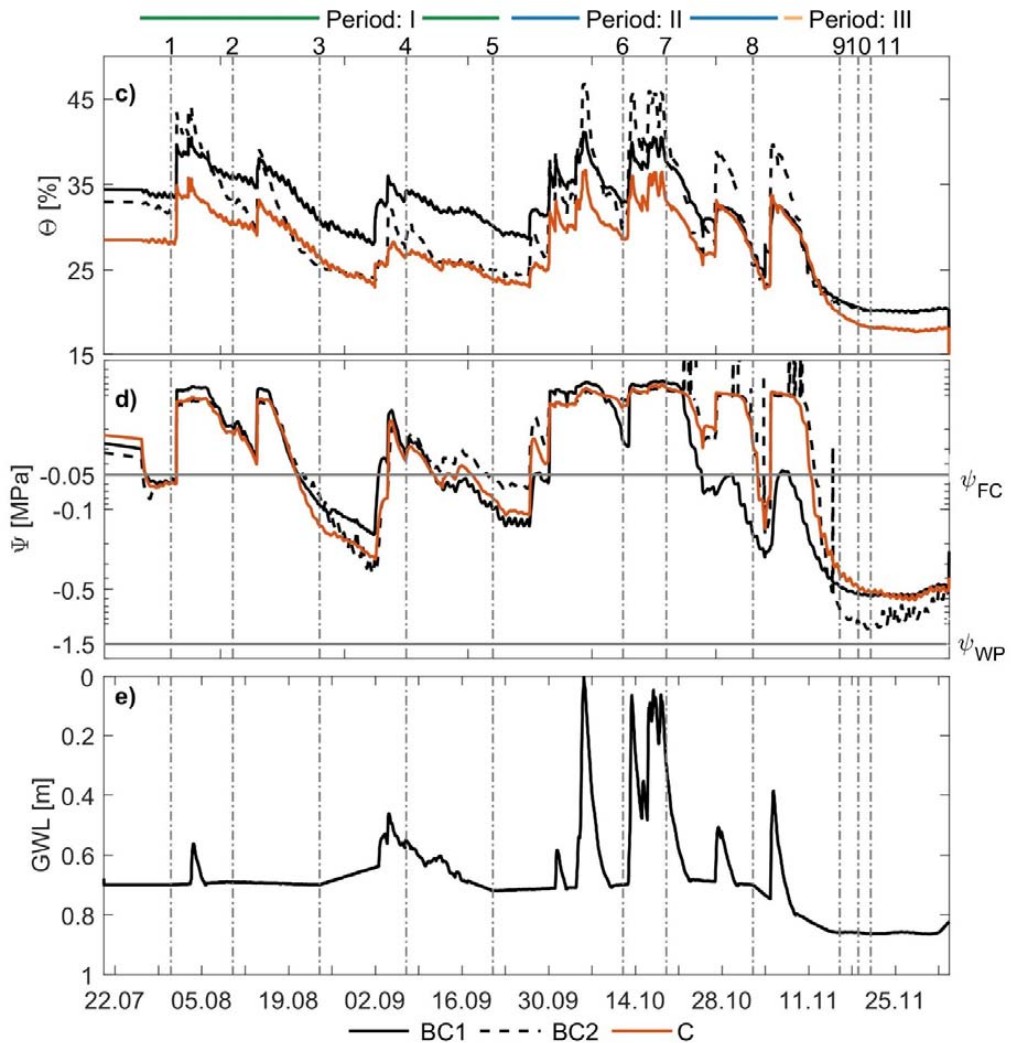

**Figure 3**    *(continued) Time series of: (c) average volumetric water content and (d) the average water potential for each treatment; (e) measured groundwater level. The different water sampling days 1-11 are indicated in each panel as vertical dashed lines and numbered on top of panel c and the date are given on the x-axis of panel e as dd.mm. Period I, II and III are indicated on the top of panel c.*

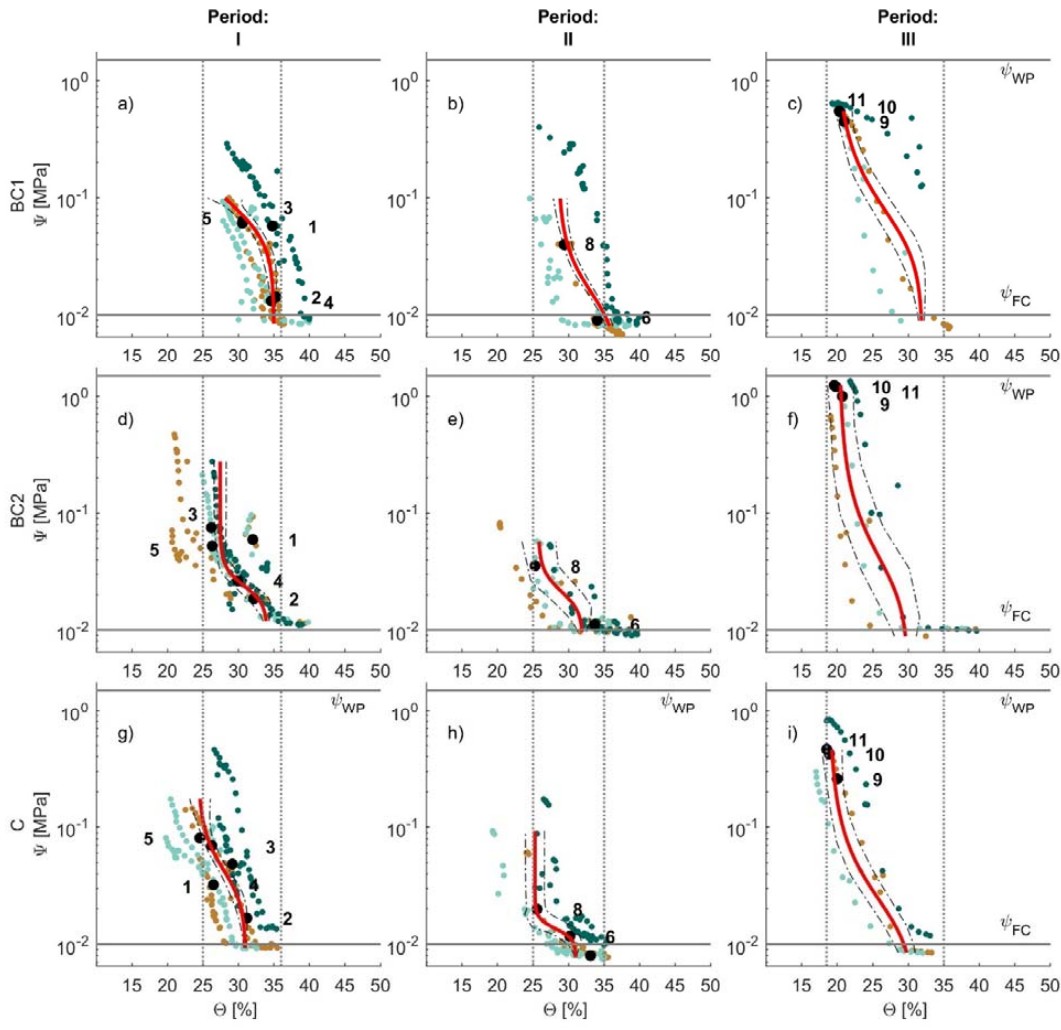

*Figure 4   The soil matric potential represented as a function of average soil water content of the different plots (colors)*
*for the treatments BC1 (a-c), BC2 (d-f) and C (g-i) and the Periods I, II and III (columns). The fitted average*
*soil water retention curves within a treatment using equation 1 (red line) including the 95% confidence interval*
*(dashed line). Black circles indicate the soil water content and soil matric potential on the sampling days*
*indicated by numbers.*



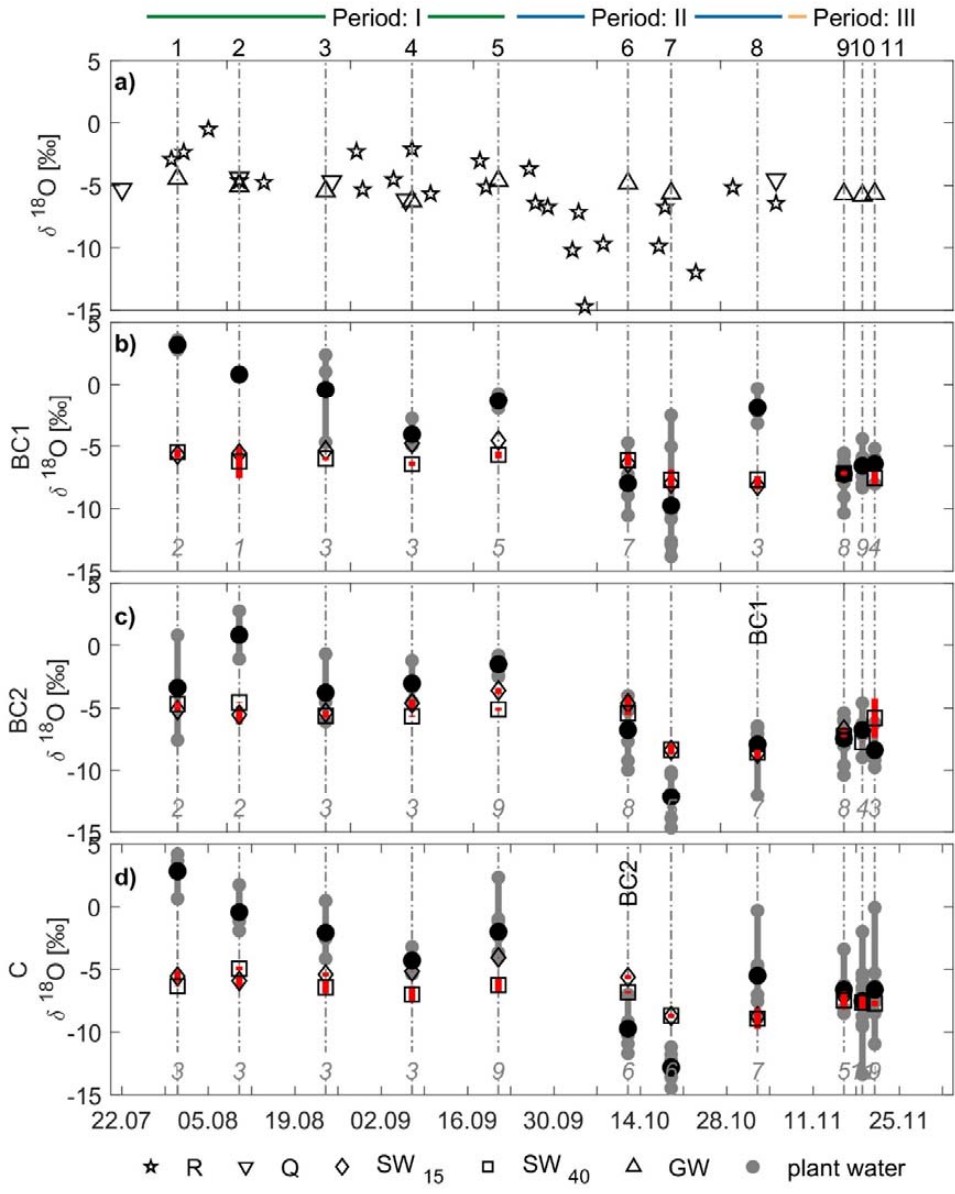

**Figure 5** *Time series of (a) δ¹⁸O in rainfall, irrigation water, ground water and (b-d) soil water sampled at 15 cm (SW₁₅) and 40 cm (SW₄₀), ranges of δ¹⁸O of SW (red line). The δ¹⁸O of plant water (grey circle) and its average (black circle) are shown for the BC1 (b), BC2 (c) and control treatment (d), for sampling days 1-11 (indicated in each panel as vertical dashed lines and numbered on top of panel a). Period I, II and III are indicated on the top of panel a. Italic numbers in panels b-d indicate the numbers of plants samples. Significant differences among the average plant water values (per treatment n>3) of each sampling day are on the vertical dashed lines as letter of the treatment e.g. BC1, BC2 or C (Tukey's honestly significant difference criterion α = 0.05).*



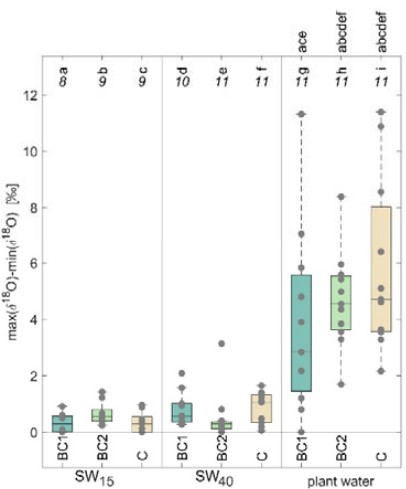
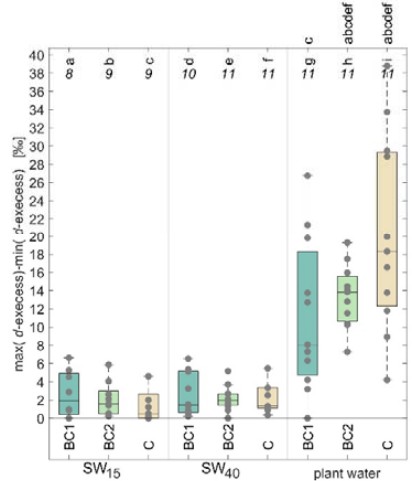

***Figure 6*** ***The variability in stable isotope composition $\delta^{18}O$ (left) and d-excess (right) expressed as range (maximum-minimum observed isotopic composition) for the soil water collected at 15 cm ($SW_{15}$) and 40 cm ($SW_{40}$) below surface, and plant water in the BC1, BC2 and control treatment. The boxes show the range of values for different sample groups (showing the median and the interquartile range, with whiskers indicating $10^{th}$ and $90^{th}$ percentiles). Circles indicate the data points. Numbers above each box indicate the number of samples available. Letters on top of each box indicate significant differences among the average values of the different groups (Tukey's honestly significant difference criterion $\alpha = 0.05$).***



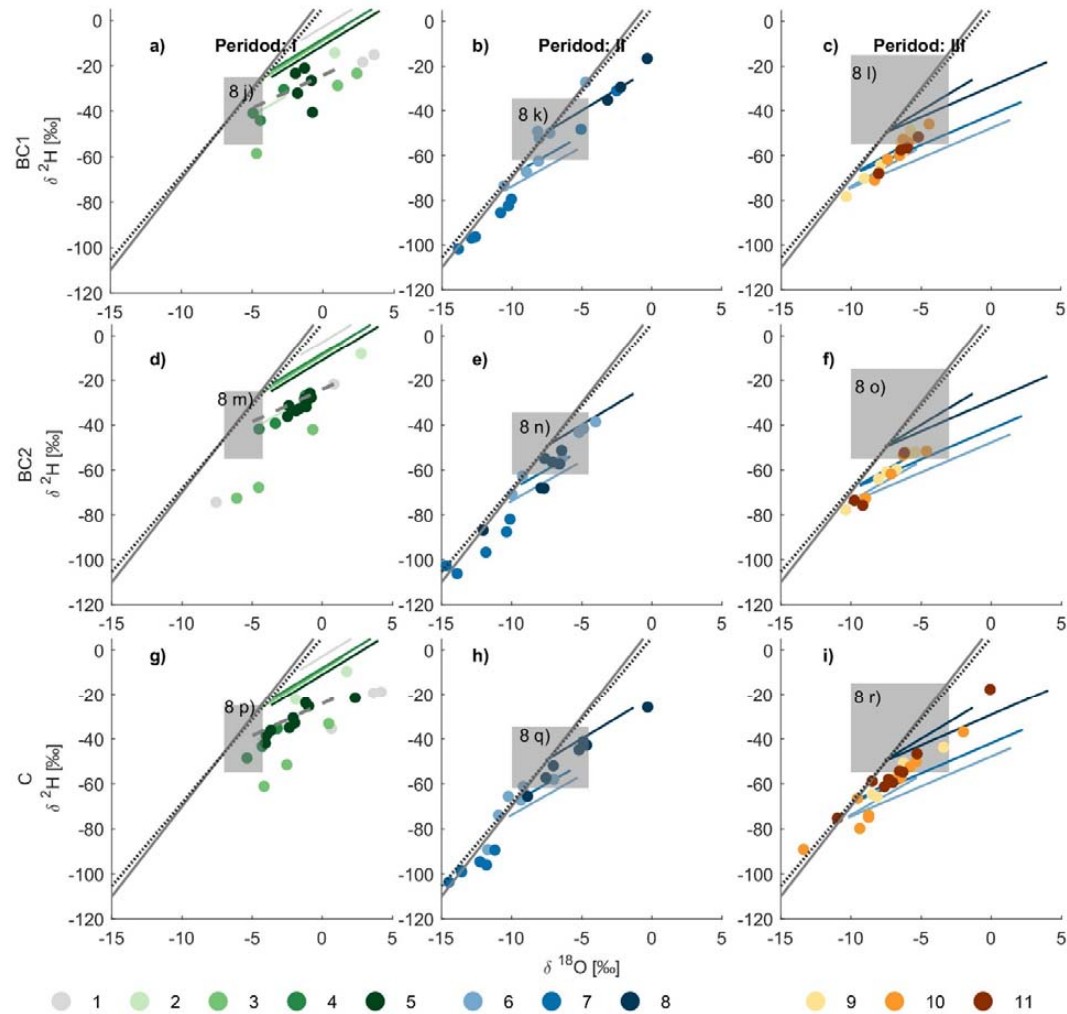


**Figure 7**  **The dual isotope space with the isotopic composition of plant water samples (circles), the calculated evaporation**
**lines of residual rainfall and sampled soil water for the treatments BC1 (a-c), BC2 (d-f) and C (g-i) and periods**
**I-III (columns). Colors indicate the different sampling days (note that lines in period III are blue because they**
**have been obtained from samples taken in period II). The local meteoric line (black dotted line) and global**
**meteoric water line (grey solid line) are indicated in all panels. The grey dashed lines (panel a, d and g) indicate**
**the evaporation line of median soil water. Isotopic compositions of irrigation, soil water and groundwater vary**
**within the grey shaded squares indicated as 8 j-8 r, and enlarged in figure 8 j-r.**

*Figure 8    The dual isotope space with the isotopic composition of irrigation (down facing triangle), soil water collected at 15 cm (SW$_{15}$, diamond) and 40 cm (SW$_{40}$, square) and groundwater (upward facing triangle). The local meteoric line (black dotted line) and global meteoric water line (grey solid line) are indicated in all panels. The different treatments BC1 (j-l), BC2 (m-o) and C (p-r) and different periods I-III (columns) indicated in grey panels of Figure 7 a-i. Colors indicate the different sampling days.*



# Appendix

*Table A1   Soil characteristics of the experimental site.*

|  | BC1 | BC2 | C |
|---|---|---|---|
| Soil (0-20 cm) texture sand/silt/clay | 34/30/36 | | |
| Infiltration capacity Wet / Dry season [mm h$^{-1}$] | 15 / 30 | 15 / 40 | 8/40 |
| pH | 6.5 | 6.3 | 6.4 |
| Ca [mol kg-1] | 11.77 | 12.43 | 11.77 |
| Mg [mol kg$^{-1}$] | 2.60 | 2.63 | 2.47 |
| K [mol kg$^{-1}$] | 0.87 | 0.97 | 0.80 |
| P [mg L$^{-1}$] | 22.3 | 29.0 | 21.6 |
| Zn [mg L$^{-1}$] | 3.2 | 3.3 | 3.1 |
| Mn [mg L$^{-1}$] | 24.0 | 30.6 | 22.0 |
| Cu [mg L$^{-1}$] | 9.3 | 11.0 | 9.6 |
| Fe [mg L$^{-1}$] | 43.00 | 57.33 | 45.00 |
| Organic C [%] | 2.29 | 2.18 | 2.16 |
| Total N [%] | 0.15 | | |



***Table A2***    ***The fitted parameters*** $\theta_r$***,*** $\alpha$ ***and*** *n the average soil water retention curves of the different treatments (BC1, BC2*
*and C) and the Periods I-III of equation 1 with the 95% confidence interval in brackets.*

| | BC1 | | | BC2 | | | C | | |
|---|---|---|---|---|---|---|---|---|---|
| | Period | | | Period | | | Period | | |
| | I | II | III | I | II | III | I | II | III |
| $\theta_r$ | 0.2 | 0.3 | 0.2 | 0.3 | 0.3 | 0.2 | 0.2 | 0.3 | 0.2 |
| | (-0.2,0.6) | (0.3,0.3) | (0.2,0.2) | (0.3,0.3) | (0.2,0.3) | (0.2,0.2) | (0.2,0.3) | (0.2,0.3) | (0.2,0.2) |
| $\alpha$ | 13 | 78 | 18 | 44 | 49 | 34 | 27 | 75 | 58 |
| | (-6.7,33) | (65,90) | (8.1,29) | (36,52) | (29,70) | (-14,81) | (20,34) | (61,88) | (30,85) |
| $n$ | 2.5 | 2.6 | 2 | 5.7 | 5.1 | 2 | 3 | 10 | 2 |
| | (-0.3,5.4) | (1.8,3.4) | (0.9,3.1) | (1.4,10) | (-0.2,12) | (0.2,3.8) | (1.2,4.8) | (-0.4,24) | (1.2,2.9) |



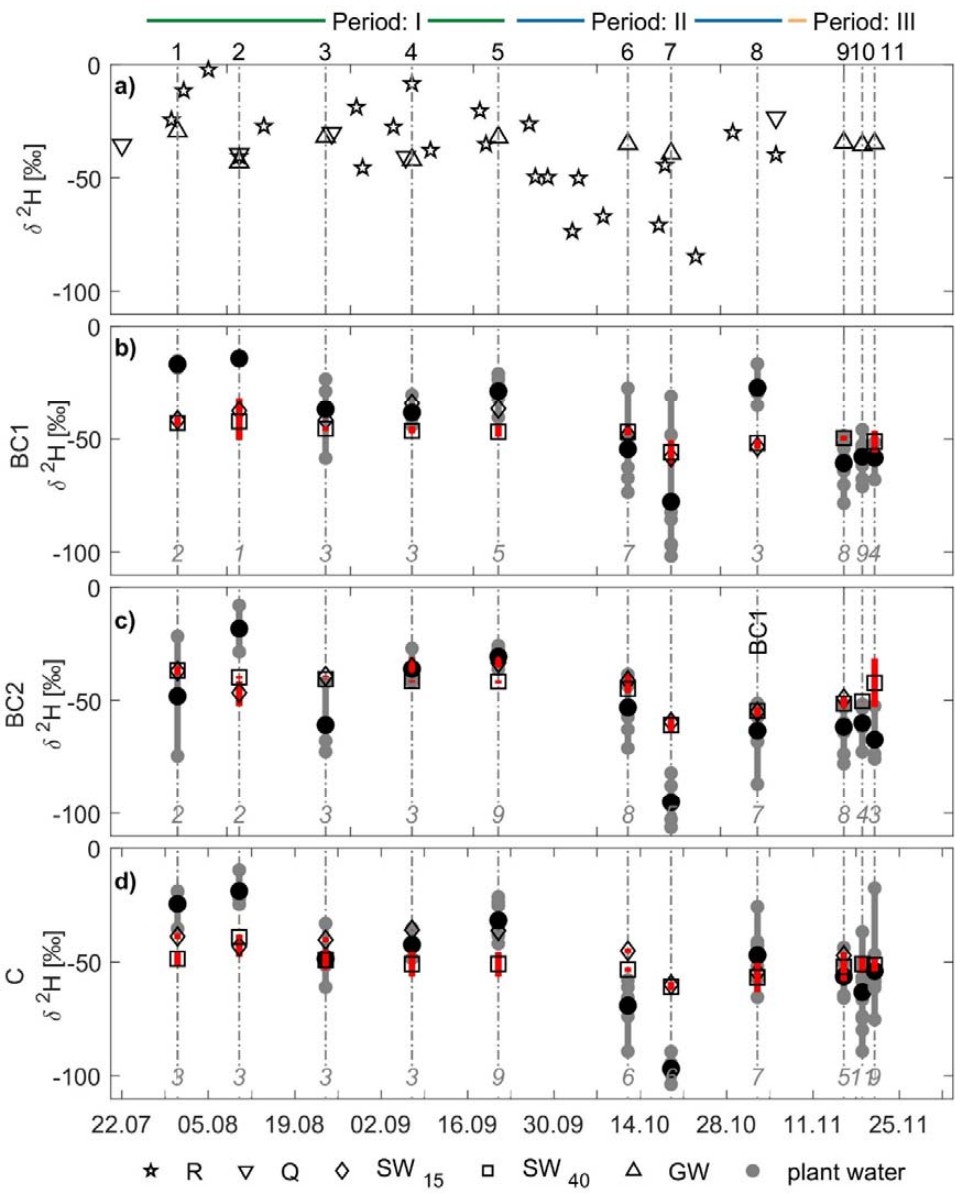

**Figure A1** *Time series of (a) δ²H in rainfall, irrigation water, ground water and (b-d) soil water sampled at 15 cm (SW15) and 40 cm (SW40), ranges of δ²H of SW (red line). The δ²H of plant water (grey circle) and its average (black circle) are shown for the BC1 (b), BC2 (c) and control treatment (d), for sampling days 1-11 (indicated in each panel as vertical dashed lines and numbered on top of panel a). Periods I, II and III are indicated on the top of panel a. Italic numbers in panels b-d indicate the numbers of plants samples. Significant differences among the average plant water values (per treatment n>3) of each sampling day are on the vertical dashed lines as letter of the treatment e.g. BC1, BC2 or C (Tukey's honestly significant difference criterion α = 0.05).*



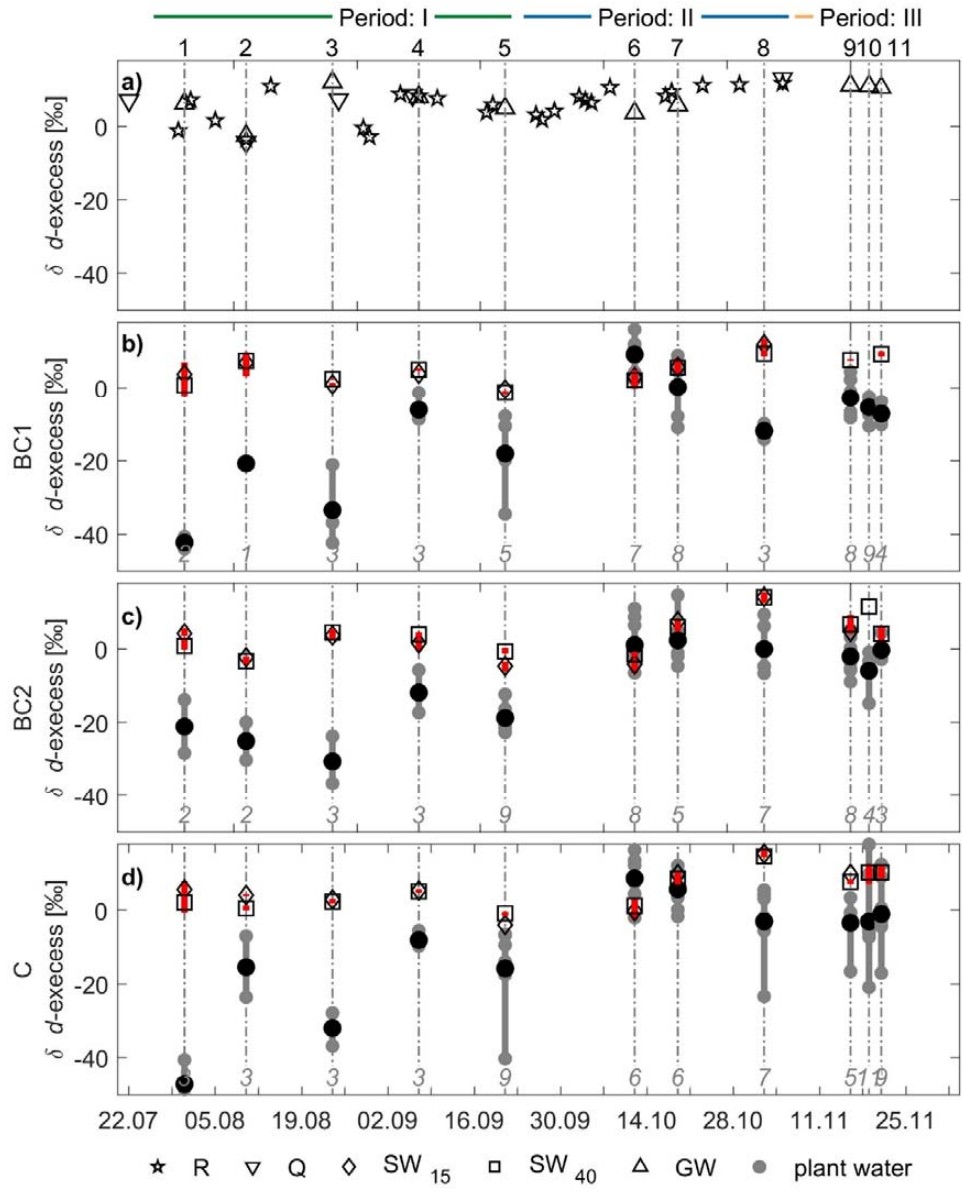

**Figure A2** *Time series of (a) d-excess in rainfall, irrigation water, ground water and (b-d) soil water sampled at 15 cm (SW$_{15}$) and 40 cm (SW$_{40}$), ranges of d-excess of SW (red line). The d-excess of plant water (grey circle) and its average (black circle) are shown for the BC1 (b), BC2 (c) and control treatment (d), for sampling days 1-11 (indicated in each panel as vertical dashed lines and numbered on top of panel a). Periods I, II and III are indicated on the top of panel a. Italic numbers in panels b-d indicate the numbers of plants samples. Significant differences among the average plant water values (per treatment n>3) of each sampling day are on the vertical dashed lines as letter of the treatment e.g. BC1, BC2 or C (Tukey's honestly significant difference criterion $\alpha$ = 0.05). The d-excess was defined as d-excess = $\delta^2H - 8 \cdot \delta^{18}O$ (Dansgaard, 1964) using data from Figure 5 and A1.*



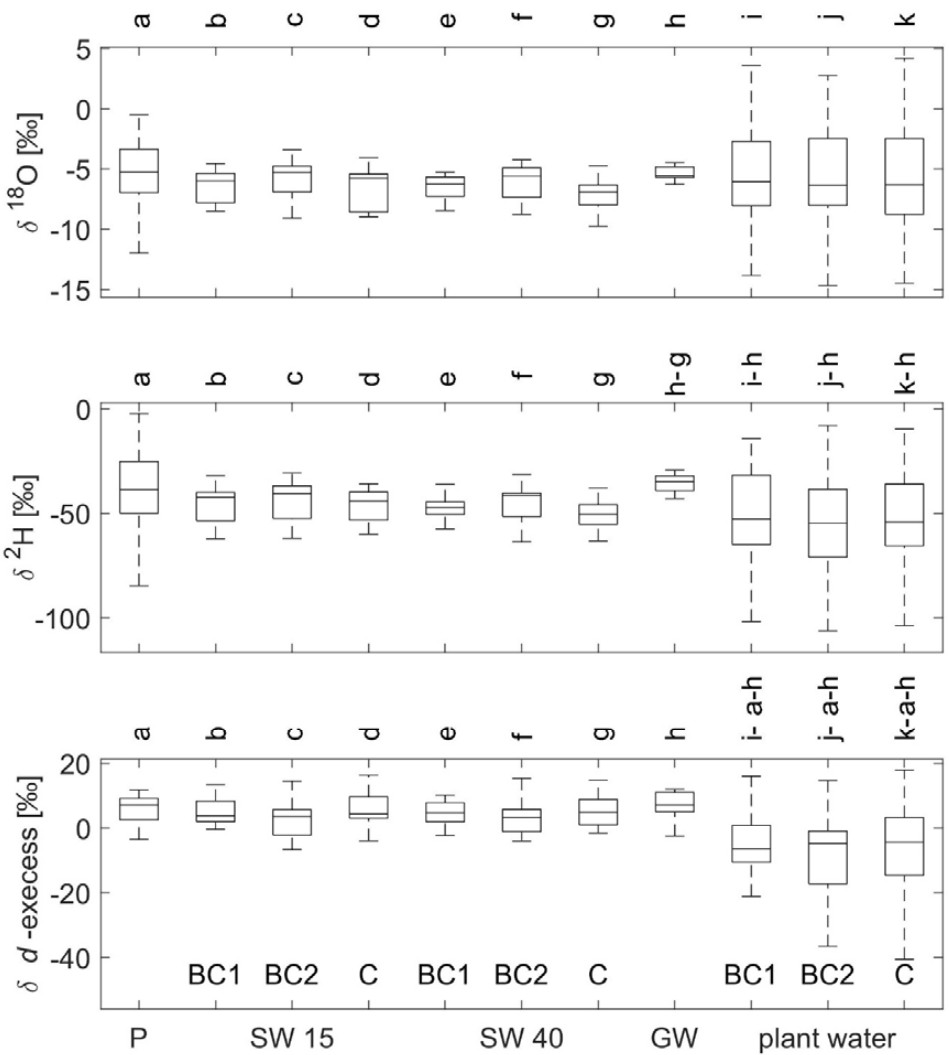

*Figure A3 The variability in stable isotope composition $\delta^{18}O$, $\delta^{2}H$ and d-excess. The x-axis indicates the sampled*
*precipitation, soil water collected at 15 cm ($SW_{15}$) and 40 cm ($SW_{40}$) below surface, groundwater and plant*
*water where BC1, BC2 and C indicate the three different treatments. The boxes show the range of values for*
*different sample groups (showing the median and the interquartile range, with whiskers indicating 10th and*
*90th percentiles). Letters on top of each box indicate significant differences among the average values of the*
*different groups (Tukey's honestly significant difference criterion α = 0.05).*