# Peer review of "Investigating the impacts of biochar on water fluxes in tropical agriculture using stable isotopes"

_Hydrology and Earth System Sciences, 2020_

## Short Comment (SC1) · 21 Aug 2020

Thank you so much for inviting me as a reviewer for Fischer and colleagues's work which use stable isotopes to investigate soil characteristics after addition of biochar and the effects on water use by rice plants. It indeed a very interesting study. The data need much hard work to get it. For me, however, there are three main concerns about this manuscript. Firstly, soil water taken by lysimeters were considered to be mobile water in the soil. This part of soil water was believed that it was not part of the water used by plants or crops (Brooks et al. 2010). The authors also did not give a very clear description for the soil water sampling in the text (start time, duration,

still sample when it rain? how to sample when it was irrigated, and so on). Differ to the trees, the sampling depth of soil water in crop filed should change following the root growth. Secondly, there are several methods to calculated the proportion of plant water from each water pools. However, the authors seems to judge it more subjectively. I suggest to use a quantitative method to calculate the proportion of various pools in the plant water such as SIAR. Finally, 1 growing season field trial on the effects of biochar amendment on soil water, water uptake of rice plants at different growth stages did not well supportÂăthe validity of the study and the observations. The effects of biochar on soil water changed over time. Minor comments: L220 applying an 800-mbar vacuum for 2 minutes. Please gave more information about soil water sampling. L221 waiting 1 hour before collecting the groundwater sample Whether 1 hour pumping will affect the ground water level? L273 method ( ET = KcÂůETref ) What is the difference for Kc or ETref between BC and C? L298 R<1 Should be "RR<1"? L554 we observed biochar amendments to create generally 2 % to 7 % higher soil 2 % to 7 % should be calculated in the Results part.

Brooks, R. J., Barnard, H. R., Coulombe, R. and McDonnell, J. J.: Ecohydrologic separation of water between trees and streams in a Mediterranean climate, Nature Geoscience, 3(2), 100–104, doi:10.1038/ngeo722, 2010.

---

## Referee Comment (RC1) · Anonymous Referee #1 · 26 Aug 2020

This work used stable isotopes to investigate soil characteristics after addition of biochar and the effects on water flux and rice water source. It is a very interesting study. The data need much hard work to get it. For me, however, there are three main concerns about this manuscript. Firstly, soil water taken by lysimeters were considered to be mobile water in the soil. This part of soil water was believed that it was not part of the water used by plants or crops (Brooks et al. 2010). For example, in dry days, the water can not be sampled by lysimeters. Is that mean there is no water could be used by plants? The authors also did not give a very clear description for the soil water sampling in the text (start time, duration, how to sample when it rain? how to sample when it was irrigated, and so on). Secondly, there are several methods to calculated

the proportion of plant water from each water pools. However, the authors seems to judge it more subjectively. I suggest to use a quantitative method to calculate the proportion of various pools in the plant water such as SIAR. After that, a comparison of water use in biochar addition or control treatment should be conducted. Finally, one growing season field trial on the effects of biochar amendment on soil water, water uptake of rice plants at different growth stages did not well support the validity of the study and the observations. The effects of biochar on soil water changed over time. Minor comments: L220 applying an 800-mbar vacuum for 2 minutes. Please gave more information about soil water sampling. L221 waiting 1 hour before collecting the groundwater sample Whether 1 hour pumping will affect the ground water level? L273 method ( ET = Kc*ETref ) What is the difference for Kc or ETref between BC and C? L298 R<1 Should be "RR<1"? L393, 397 It seems that the minimum value of plant water of 18O was smaller than the soil water's? How to explain this? L409, 411-412 Please use the same order for the data. From low to high? e.g. -3.7 ‰ to -12.7 ‰ or -12.7 ‰ to -3.7 ‰ L527 there are two access? L554 we observed biochar amendments to create generally 2 % to 7 % higher soil. 2 % to 7 % should be calculated in the Results part. Brooks, R. J., Barnard, H. R., Coulombe, R. and McDonnell, J. J.: Ecohydrologic separation of water between trees and streams in a Mediterranean climate, Nature Geoscience, 3(2), 100–104, doi:10.1038/ngeo722, 2010.

---

## Referee Comment (RC2) · Anonymous Referee #2 · 16 Sep 2020

The authors examined the water sources of rice crop plants in soils amended with two different biochar types in comparison to non-amended soils (control treatment) in a seasonally dry tropical region in Costa Rica. For this, they used stable isotope signatures of precipitation, irrigated water, soil-lysimeter water, ground water and plant stem in combination with soil moisture and matric potential and climate data. Overall, their findings showed increases of plant available water in amended soils and, across treatment plots, the stable water isotope composition of plant water showed that the rice plants preferentially used water from the first 20 cm of soil.

The main objective of this study was to investigate plant water sources. As background

information provided (L113-115), the study of Shen et. (2015) showed that flooded rice consumed soil water from 0-15 cm deep, while the study of Mahindawansha et al. (2018) found that upland rice in dry conditions mostly consumed soil water from the first 50 cm of depth. Based on these findings, the authors hypothesized that amending biochar into the top 10-30 cm of the soil, could increase soil water storage and rice plants be able to use water from different soil water pools in comparison to plants grown in non-amended soils (L113-125) across the different time periods studied.

First, I should say that after the work of Brooks et al. (2020), countless studies across regions (including tropical wet environments; see Goldsmith et al. 2012; Muñoz-Villers et al. 2018) and revisions have showed that plants use evaporatively fractionated soil water (Sprenger et al., 2016; Sprenger and Allen 2020). The general finding is that plant water is isotopically similar to bulk soil water but not to low suction‐lysimeter water, implying that roots are generally located in less conductive (mobile) pores where water tends to travel more slowly and can reside for longer times.

In the present study, soil water samples representing the soil water pool for plants, were only collected using low suction (80 kPa) lysimeters. Bulk soil samples were a key part of the experiment but they are missing here. This methodological issue, is perhaps the largest flaws of the research and I do not see a way to get out of it, based on all the evidence published over the last decade.

In addition, your soils are dominated by clay content (Table A1) which is very well known for its very fine particle structure making very difficult to empty the water from such smaller pores using low soil tension lysimeters. This is other reason why the authors should have collected and used bulk soil water isotope ratios (from cryogenic vacuum distillation) as the representative soil water source for plants. I also observed that your samplings 9,10 and 11 corresponding to Period III (Figure 4), were characterized by low soil water contents held at very high tensions (close to PWP conditions). Hence, the soil water collected with soil lysimeters was not "seeing" the water that plants were extracted during this dry period. This situation is particularly observed for the biochar

amended treatments. Therefore, the research question 2 cannot be answered.

I have made some other important comments that the authors can also consider when preparing other articles around these topics:

1) The use of mixing models to quantify the relative contributions of the different plant water sources, instead of reporting the results in a visual graphical and/or descriptive way only.

2) The construction of dual isotope space figures in which the plant and the different water sources are plotted together. In this way, it is easier the assess the isotope information per sampling period and seasons (Figure 7 and 8).

3) Both water isotopes ($\delta$2H and $\delta$18O) were determined for the plant and potential water sources, however, the results were only elaborated around 18O. I would suggest you to describe both isotope ratios.

References

Brooks, J. R., Barnard, H. R., Coulombe, R., & McDonnell, J. J. (2010). Ecohydrologic separation of water between trees and streams in a Mediterranean climate. Nature Geoscience, 3(2), 100–104.

Goldsmith GR, Muñoz-Villers LE, Holwerda F, McDonnell JJ, Asbjornsen H, Dawson TE. Stable isotopes reveal linkages among ecohydrological processes in a seasonally dry tropical montane cloud forest. Ecohydrol 5:779–790, 2012.

Muñoz-Villers, L.E., Holwerda, F., Alvarado-Barrientos, M.S., Geissert, D.R., and Dawson, T.E.: Reduced dry season transpiration is coupled with shallow soil water use in tropical montane forest trees, Oecologia, 188, 303–317, 2018.

Sprenger, M., Leistert, H., Gimbel, K., & Weiler, M. (2016). Illuminating hydrological processes at the soil‐vegetation‐atmosphere interface with water stable isotopes. Reviews of Geophysics, 54, 674–704.

[Figure]

Sprenger, M., and Allen, S. T. (2020). What ecohydrologic separation is and where we can go with it. Water Resources Research, 56, e2020WR027238.

---

## Author Comment (AC1) · 8 Jan 2021

**Hess-2020-404     Author reply on comment from Anonymous Reviewer #1**

*This work used stable isotopes to investigate soil characteristics after addition of biochar and the effects on water flux and rice water source. It is a very interesting study. The data need much hard work to get it.*

We thank Reviewer #1 for the constructive feedback and are happy the reviewer appreciated our work to collect the data for this manuscript. Please find below our responses to the individual comments and suggestions (reviewer #1 comments in blue font, with our response in black font).

*For me, however, there are three main concerns about this manuscript. Firstly, soil water taken by lysimeters were considered to be mobile water in the soil. This part of soil water was believed that it was not part of the water used by plants or crops (Brooks et al. 2010). For example, in dry days, the water can not be sampled by lysimeters. Is that mean there is no water could be used by plants? …*

As described in L103 the soil water can consist of different pools of water: "…plants use mobile vs. immobile soil water pools (Brooks et al., 2010)…". To infer which water plants use, lysimeters are commonly used to sample mobile soil water (Sprenger et al., 2015). Instead to access the tighter bound and more fractioned soil water with matric potential <-0.1 MPa, the immobile soil water, it is necessary to use e.g. the cryogenic vacuum method (Sprenger et al., 2015). Therefore, we agree with the reviewer that all soil water collected with lysimeters should be considered mobile soil water. To provide better context and avoid misrepresentation we adjusted the wording where appropriate to mobile soil water when we refer to samples collected using lysimeters and specify in L101:

> To identify which water stores are available to vegetation, various potential water sources are collected -e.g., rain (Fischer et al., 2019; Prechsl et al., 2014), mobile soil water (lysimeters) or immobile water (tighter bound and more fractioned soil water with matric potential <-0.1 MPa using e.g. the cryogenic vacuum) method (Sprenger et al., 2015) and groundwater (Beyer et al., 2016). The different water samples are then analyzed for their stable isotope composition.

During the Period I consisting of dry and wet spells and the wet Period II we were able to extract with the lysimeters mobile soil water samples. Instead in Period III, it is correct that not all lysimeters could extract water. To have information on the full spectrum of the mobile and immobile soil water we did collect soil cores for bulk soil water extraction, especially in low matric potential situations as in Period III. On average, however, we could extract less than 0.1 ml of water per soil sample using cryogenic vacuum extraction. This was too little water for pipetting and analysis in the LIS-autosampler setup. As a result, we could not gain information on the immobile water isotopic composition.

We have included the following text in the method section of our manuscript:

After Old L224:

> In addition to the lysimeters, soil samples for bulk soil water extraction and subsequent stable isotope analysis in were collected randomly in all plots from a depth of ~10 cm on 7 out of 11 sampling days. In order to not disturb the rice plants, instead of an auger, a steel rod 50 cm in length with a 2 cm diameter was pushed 10 cm into the soil. After removing 5 cm of the topsoil, the soil sample was collected (~5 cm ∅2cm). The sample was then placed in a double resealable zipper storage bag. To minimize post-sampling evaporation, the storage bags were directly placed in a cooler with ice. All soil samples were stored in the laboratory freezer (-80 °C) before extracting the soil water for isotopic analysis.

And old L235-246:

Plant water was extracted from the stem (xylem water) of the different rice plants to infer which sources of water the rice plants used. We used the cryogenic vacuum extraction technique described by Koeniger et al., (2011) to extract the plant and bulk soil water for stable isotope analysis. The method uses a heated vial and a cold trap vial (Exetainer® vial with standard cap and rubber septum, Labco Ltd, Lampeter, United Kingdom) connected with stainless-steel capillary tubing. About 3 g of plant material from the rice stem was placed in the heated vial before the system was evacuated to 85 kPa with a vacuum hand pump (Mityvac). The heated vial was heated for 1 hour at 100°C using a test tube heater (HI839800 COD Test Tube Heater; Hanna instruments) while the cold trap vial rested in a Dewar flask containing liquid nitrogen at about -196°C. After the extraction was stopped, the cold trap vial was sealed with Parafilm and left to thaw. After thawing, the extracted liquid water was pipetted into 2 ml vials (32 x 11.6 mm screw neck vials with cap and PTFE/silicone/PTFE septa) and stored refrigerated (5 °C) until stable isotope analysis. The plant root and bulk soil water was extracted in the same manner as the xylem water using the cryogenic vacuum extraction technique but with extraction time longer than 3 hours. On average 86±5 % plant water and soil water were extracted. However, we extracted less than 0.1 ml of water per soil sample for the bulk soil water and less than 0.1 ml of water per root sample for the root water which were too small volumes for pipetting and the LIS-autosampler setup.

*… Secondly, there are several methods to calculated the proportion of plant water from each water pools. However, the authors seems to judge it more subjectively. I suggest to use a quantitative method to calculate the proportion of various pools in the plant water such as SIAR. After that, a comparison of water use in biochar addition or control treatment should be conducted.*

We felt that due to the lack of bulk soil water information it would be rather uncertain and potentially erroneous to calculate the fraction of different plant water sources, which could lead us to over interpret our data. However, we agree that a better job needs to be done quantifying potential plant sources to avoid being overly qualitative. Thus, we have quantified, based on the available isotopic data, the fraction of different plant water sources to better compare the plant water use in biochar amended treatments with the control treatment.

From the descriptive analysis of the collected isotope data, we developed a three-component mixing model to help identify plant water sources and allow for comparison between treatment and control. We identified soil water from recent rainfall as a first potential plant water source (Figure 5-8 and A4). Seeing the rather small range (<1 ‰ $\delta^{18}$O and <5‰$\delta^2$H) and partly overlapping isotopic composition of mobile soil water and ground water (Figure 5, 6, 8 and A4), we chose the median isotopic composition of soil water collected at -15 cm, the mobile water, as a potential second plant water source. In addition, we observe in Figure 7 that the plant water samples drifted in time from the GMWL along the theoretical evaporation line of the soil water from antecedent rainfall. From this, we assume that the end point of the theoretical evaporation line can serve as proxy of the third potential water source if plants access tighter bound soil water - the immobile water. These three end-members were used in a

simple three-component mixing model to explore and support which sources of water the rice plants potentially consumed during the different sampling days.

The end-member mixing model will be presented in the method section after L328 as:

In addition, in the different treatments the potential plant water use of the rice plants was quantified using mixing model. Mixing models are powerful tools to estimate the plant water sources (Layman et al., 2012; Rothfuss and Javaux, 2017). However, applying Bayesian mixing models would not decrease the uncertainty in potential plant water sources due to the missing bulk soil water isotope data due to the little amount of cryogenically extracted bulk soil water. Instead, a simple three end-member mixing model was used:

$$O_{PW} = f_R O_R + f_{SI} O_{SI} + f_{SM} O_{SM} \tag{9}$$

$$D_{PW} = f_R D_R + f_{SI} D_{SI} + f_{SM} D_{SM} \tag{10}$$

$$1 = f_R + f_{SI} + f_{SM} \tag{11}$$

$$f_R = \frac{O_{PW} D_{SM} - O_{PW} D_{SI} + O_{SM} D_{SI} - O_{SM} D_{PW} + O_{SI} D_{PW} - O_{SI} D_{SM}}{O_R D_{SM} - O_R D_{SI} + O_{SM} D_{SI} - O_{SM} D_R + O_{SI} O_R - O_{SI} D_{SM}} \tag{12}$$

$$f_{SI} = \frac{O_{PW} D_{SI} - O_{PW} D_R + O_R D_{PW} - O_R D_{SI} + O_{SI} D_R - O_{SI} D_{PW}}{O_R D_{SM} - O_R D_{SI} + D_{SM} D_{SI} - D_{SM} O_R + O_{SI} D_R - O_{SI} D_{SM}} \tag{13}$$

$$f_{SM} = \frac{O_{PW} D_R - O_{PW} D_{SM} + O_R D_{SM} - O_R D_{PW} + O_{SM} D_{PW} - O_{SM} D_R}{O_R D_{SM} - O_R D_{SI} + O_{SM} D_{SI} - O_{SM} D_R + O_{SI} D_R - O_{SI} D_{SM}} \tag{14}$$

where $f_R$, $f_{SI}$, $f_{SM}$ [-] represents the fraction of plant water source of soil water from antecedent rainfall, immobile and mobile soil water while $O$ and $D$ indicate $\delta^{18}O$ and $\delta^2H$ [‰] respectively where subscript PW, R, SI and SM indicate the plant water, soil water with signature from antecedent rainwater, immobile and mobile soil water respectively. Seeing the limitation of the missing immobile water the three end-member mixing model was rather used as an explorative approach to support the more descriptive data analysis to explain potential plant water use of the rice plants in the different treatments.

The results will be presented after the results of the dual isotope space in L440 as:

To explain potential plant water use of the rice plants in the different treatments a simple end-member mixing model was used (equation 9-14). We identified three different end-members from the descriptive analysis of the collected isotope data (Figure 5-8 and A4). The first end-member, for sampling day 1, 3-5 was defined as the volume weighted isotope composition of rainfall collected in the two-week period before a given plant water sampling. The isotopic composition of plant water during sampling day 2 was closest to the median isotopic composition of rainfall of period 1 and therefore used as end-member for period 2. The isotopic composition of plant water during sampling day 9-11 was closest to the median isotopic composition of rainfall of period 6 and therefore used as end-member for period 9-11. Instead,

the isotopic composition of plant water during sampling 5 was closest the isotopic composition of rainfall of the high rainfall event of 5 October (P=93 mm d$^{-1}$) and therefore used as end-member "soil water from antecedent rainfall" for sampling day 6-8. Seeing the rather small range (<1 ‰ $\delta^{18}$O and <5‰ $\delta^2$H) and partly overlapping isotopic composition of mobile soil water and ground water (Figure 5, 6, 8 and A4) we chose the median isotopic composition of the mobile soil water collected at -15 cm as potential second plant water sources. Finally, we observed that the plant water samples drifted in time from the GMWL along the theoretical evaporation line of the soil water from antecedent rainfall (Figure 7). From this, we assume that the end point of the theoretical evaporation line can serve as proxy of the third potential water source if plants access tighter bound soil water -the immobile water.

From the mixing model, results show that the fraction of the three plant water sources of the different treatments were variable for each sampling day (Figure 9). In cases where plant water sources contributed more than 100% or negatively the calculate the fraction was excluded as these cases clearly violate the base assumptions of mixing models. Therefore, the number of times it was possible to calculate the fraction of different plant water sources, out of the total of plant water samples of a treatment on a sampling day, differed for each treatment and sampling day.

During Period I, few plant water samples were available but generally the plant consumption of immobile water decreased with a simultaneous increase in soil water from antecedent rainfall and mobile soil water (Figure 9). In Period II plants consumed soil water from antecedent rainfall and mobile soil water (together ~80%) and to a lesser extent immobile water. During wetter sampling day 7 the plants consumed dominantly soil water from antecedent rainfall (up to 80%) and to a lesser extent immobile water and mobile soil water. Instead on sampling day 8 plants consumed less soil water from antecedent rainfall and mobile soil water (less than ~30%) while dominantly immobile water. In Period III the plants consumed largely soil water from antecedent rainfall (~ 60%) and equally immobile and mobile soil water (each ~ 20%). In none of the sampling days a significant difference was observed between the different treatments for the different sampling days (Tukey's honestly significant difference criterion α = 0.05).

[Figure]

*Figure 9  The fraction of plant water source rainfall (top), soil water from antecedent rainfall (middle) and the mobile soil water (bottom) indicated for each plant water sample (circle), the median fraction of the treatment (red line) and box plots (n>5) for the different treatments BC1, BC2 and C (indicated with grey letters) for the different sampling days 1-11. The boxes show the range of values for different sample groups (showing the median and the interquartile range, with whiskers indicating 10th and 90th percentiles). Italic grey numbers on the top panel indicate the number of times it was possible to calculate the fraction of different plant water sources out of the total of plant water samples of a treatment on a sampling day.*

[Figure]

*Figure A4 The stable isotope composition δ¹⁸O (left) and δ²H (right) for precipitation (P), immobile water (SI), mobile water (SM) and plant water of the BC1, BC2 and control treatment for the sampling days 1-6 (rows). Circles indicate the data points. The boxes show the range of values for different sample groups (showing the median and the interquartile range, with whiskers indicating 10ᵗʰ and 90ᵗʰ percentiles). The red bar indicates the used end-member used in the mixing model (P-SM) and the median value for BC1, BC2 and C.*

[Figure]

*Figure A4 (continue) The stable isotope composition δ¹⁸O (left) and δ²H (right) for precipitation (P), immobile water (SI), mobile water (SM) and plant water of the BC1, BC2 and control treatment for the sampling days 7-11 (rows). Circles indicate the data points. The boxes show the range of values for different sample groups (showing the median and the interquartile range, with whiskers indicating 10th and 90th percentiles). The red bar indicates the used end-member used in the mixing model (P-SM) and the median value for BC1, BC2 and C.*

The results and limitation were discussed in section 5.3 Temporally variable plant water sources after L510:

> The rice plants in this study had different water sources available during different periods of the experiment, but what water did they consume?
>
> Mixing model results in Period I (Figure 9), indicated that it is likely that the plants consumed dominantly immobile. This is consistent with results observed in previous studies using stable water isotopes to map out plant water sources (Brooks et al., 2010; Penna et al., 2020; Sprenger et al., 2016). This interpretation of plant water composition is supported by plant water samples falling along the theoretical evaporation lines estimating how soil water would evolve isotopically due to evaporation (Figure 7). Therefore, it is likely that during Period I, the young rice plants (with shallow root system <20 cm as reported by Mahindawansha et al. 2018) consumed the immobile (Figure 7 and 9) which was not sampled with the lysimeters at 15 cm and 40 cm below the surface. We could, unfortunately, cannot confirm this as we could not extract enough bulk soil water for isotopic analysis.

During Period II, plants grew to their maximum heights with roots reaching deeper soil layers (depth >60 cm as reported by Mahindawansha et al. 2018). This means that the rice plants, similar to larger vegetation (Allen et al., 2019), would have had access to deeper and more-stable pools of water with a distinct lower *d-excess* signature. However, the isotopic composition of plant water during this period followed the GMWL (Figure 7 b, e and h) and mixing model results (Figure 9), indicating that plants consumed largely shallow soil water from recent rainfall.

In Period III, it became increasingly difficult to extract water from lysimeters at 15 cm below the surface and the isotopic composition of plant water drifted from the GMWL, falling along the theoretical evaporation line of residual rainfall falling in Period II (Figure 7 l, o and r) which is supported by the mixing model results (Figure 9). With the experiment being held in the tropics and based on the findings from Amin et al (2020) one would expect that the rice plants with their longer roots would accessed the more stable and older water stores in deeper subsurface zones below 60 cm. Instead, the rice plants in the different treatments preferably consumed the temporally variable and "newer" surface near soil water from recent rainfall similarly to what has been documented in grasslands (Bachmann et al., 2015) and catchments (e.g. van der Velde et al., 2015).

As previously stated, mixing models are only as good as the available data. Despite the draw back and source of uncertainty in the simple mixing model due to the missing bulk soil water isotopic composition, the results were still useful as explorative tool to support the more qualitative data analysis (Figure 5-8 and A4). Our results provided a first insight that plants water sources were largely variable within a treatment but no difference between biochar and control treatments could be observed. Seeing the large uncertainty and variability of plant water sources demonstrates the need of further and more detailed research of plant water use in biochar amendments. By performing more detailed isotopic experiments (Beyer et al., 2016), higher temporal resolution sampling of plant water (Marshall et al., 2020; Volkmann et al., 2016) and spatiotemporal soil water (Sprenger et al., 2015) or including interception, transpiration and atmospheric processes into the experimental analysis (Jiménez-Rodríguez et al., 2020) which would allow to not only distinguish whether the rice plants prefer mobile or . immobile water (Berry et al., 2018; Brooks et al., 2010; McDonnell, 2014; Muñoz-Villers et al., 2020) but also to more accurately quantify the fraction of water sources. Next to the aforementioned vertical processes, the lateral water fluxes (Sprenger and Allen, 2020) need to be considered to assess the field-scale responses to biochar amendments (Fischer et al., 2018) which would allow to better constrain the dominant ecohydrological process as e.g. Muñoz-Villers et al.(2020). Furthermore, when mixing biochar in the top soil, a multi-layer soil profile is created and based on studies in natural catchments, e.g. Penna et al. (2018) or Sprenger et al. (2016), these different layers could store not only different quantities of water but also water characterized by different ages. In addition, Blanco-Canqui (2017) discusses how biochar can age thereby altering the physical and chemical characteristics of biochar and soil in time. The long-term effect of biochar soil amendments in tropical agroecosystems would also need to be considered especially to understand and identify which sources of water the rice plants consume from year to year.

Although these more detailed short and long-term analyses were beyond the scope of this initial investigation our results from one growing season indicate that rice plants growing in biochar amended soils not only had access to more water (Figure 4) but also had a more stable source of green water (i.e., soil moisture from rainfall) and thus could withstand dry spells seven days longer (Figure 3). Regardless of the potential advantages, as stated by Fischer et al. (2018), it must be noted that biochar as water management tool does not adhere to a one size fits all approach but needs fine tuning in accordance with climate, site and plant characteristics to obtain stable and optimal yields.

*The authors also did not give a very clear description for the soil water sampling in the text (start time, duration, how to sample when it rain? how to sample when it was irrigated, and so on).*

We thank Reviewer 1 for this suggestion. As specified in the manuscript, the different sampling campaigns were held when there was no rainfall (sampling was conducted just before or after rainfall). The experiment had an emphasis on rain fed agriculture. The different plots were only irrigated as specified on 22 July (before the experiment) and on 25 August (in between two sampling days). These data were also presented in Figure 8. Although we specified the sampling time of the rice plants, we agree with the reviewer that the soil water sampling was only briefly described. We will clarify this in the new version of our manuscript and include after L218:

> Soil water and groundwater samples were collected during dry days or when there was no rainfall (sampling was conducted just before or after rainfall) approximately biweekly after plant germination from 31 July 2018 until the harvest day on 21 November 2018, resulting in 11 sampling days. At approximately 9H00 on each sampling day, on each lysimeter a vacuum of 800-mbar was applied for one hour. After releasing the vacuum of each lysimeter, on average 100 ml of soil water was collected.

In addition, instead only stating that water samples were collected we specify in L221:

> *The different soil and groundwater samples were collected in 30 ml PE bottles,...*

*Finally, one growing season field trial on the effects of biochar amendment on soil water, water uptake of rice plants at different growth stages did not well support the validity of the study and the observations. The effects of biochar on soil water changed over time.*

Working with one season could be interpreted as weakness to generalize findings from our study in comparison to pot, laboratory and greenhouse experiments where it is "easier" to control all variables and boundary conditions. However, as stated in the manuscript in L80: "...*laboratory and pot experiments unable to mimic the variety of processes occurring in agroecosystems at field scale (Agegnehu et al., 2017; Blanco-Canqui, 2017; Zhang et al., 2016)*". However, we belief that even short-term studies of one growing season are useful to learn about the effect of biochar amendments in agroecosystems indicating: 1) biochar increases the soil water content and 2) rice plants consume preferably soil water from the top 20 cm soil layer.

We will highlight the short-term character and put these in context we included in L544:

> In addition, Blanco-Canqui (2017) discusses how biochar can age thereby altering the physical and chemical characteristics of biochar and soil in time. The long-term effect of biochar soil

amendments in tropical agroecosystems would also need to be considered especially to understand and identify which sources of water the rice plants consume from year to year.

Although these more detailed short and long-term analyses were beyond the scope of this initial investigation our results from one growing season indicate that rice plants growing in biochar amended soils not only had access to more water (Figure 4) but also had a more stable source of green water (i.e., soil moisture from rainfall) and thus could withstand dry spells seven days longer (Figure 3). Regardless of the potential advantages, as stated by Fischer et al. (2018), it must be noted that biochar as water management tool does not adhere to a one size fits all approach but needs fine tuning in accordance with climate, site and plant characteristics to obtain stable and optimal yields.

*Minor comments:*

*L220 applying an 800-mbar vacuum for 2 minutes. Please gave more information about soil water sampling.*

As stated above, we will include more information on how we performed the soil water sampling.

*L221 waiting 1 hour before collecting the groundwater sample Whether 1 hour pumping will affect the ground water level?*

Groundwater in the groundwater well can be subject to evaporation and thus fractionated. Therefore, it is important to purge the groundwater well before collecting the water sample. The depth of the groundwater well was 90 cm consisting of a PVC pipe ∅5.1 cm. From the groundwater level and dimension the volume of water was general less than 1 L. By purging the well and **waiting one hour** (not pumping for one hour) before collecting the sample the groundwater is minimally affected over a larger distance.

We noticed that we did not include the dimension of the groundwater well but will include this in the revised version.

*L273 method ( ET = Kc\*ETref ) What is the difference for Kc or ETref between BC and C?*

In the presented manuscript we used same Kc and $ET_{ref}$ values for both BC and C treatments. $ET_{ref}$ represents the evaporation demand from the atmosphere from a standardized vegetated surface and it is exclusively driven by atmospheric meteorological conditions which are homogeneous among treatments (Allen et al., 1998). The Kc factor serves as an aggregation of the physical and physiological differences between the studied crop and the standardized vegetated surface (Allen et al., 1998). Different levels of simplifications have been accepted in the literature when applying Kc values to specific crops (e.g. from assuming same values for different varieties of the same crop or same crops grown in completely different regions, to using standardized global tabulated values of Kc for each crop.) In order to increase the reliability of our ET estimates, we decided to use region-specific empirically (eddy covariance) derived Kc values for our rice crop which are quite rare and much more reliable than globally tabulated Kc values. Although we recognize, BC treatments might have affected to certain degree some crop physiological conditions, the comparison of the evaporative fluxes in biochar and non-amended crop plots were beyond the scope of this study. The complexities associated to the accurate estimation of the physical and physiological factors controlling the values represented

by Kc are the subject of a different research topic. Jin et al. (in review) estimated sensible heat fluxes and net radiation from thermal and optical remote sensing data in period III. There were not significant differences in sensible heat fluxes among BC and C treatments in the last part of the season. The differences in latent heat flux between BC1 and BC2 with respect to the control were of 1% and 4 % respectively. This supports the assumption of using similar Kc across treatments.

*L298 R<1 Should be "RR<1"?*

Correct, the single R should be double indicating the response ratio RR. We will correct this as RR.

*L393, 397 It seems that the minimum value of plant water of 18O was smaller than the soil water's? How to explain this? L409,*

Indeed, the $\delta^{18}O$ of plant water on sampling day 9 was lower than the $\delta^{18}O$ of soil water. However, the $\delta^{18}O$ of plant water was similar to the $\delta^{18}O$ of rainfall on sampling day 6 (Figure 7h) indicating that it is likely that rice plants consumed antecedent rainwater. We will include comments on this in our revised version of the manuscript.

*411-412 Please use the same order for the data. From low to high? e.g. -3.7 ‰ to -12.7 ‰ or -12.7 ‰ to -3.7 ‰*

We agree that isotopes should be logically written from negative to positive. However, in L411 we emphasised the change in isotopic composition of rainfall in Figure 5a which changed from -3.7 ‰ to -12.7 ‰ after sampling day 7. Therefore, it might be less confusing to keep the current sentence.

*L527 there are two access?*

Thank you for pointing this out. We removed the second access.

*L554 we observed biochar amendments to create generally 2 % to 7 % higher soil. 2 % to 7 % should be calculated in the Results part.*

We agree with the reviewer that the percentages were not clear. The 7% and 2% indicate the increased soil moisture in biochar amended soils in Period I and III respectively resulting from the minimum soil moisture observed in the different treatments (Figure 4). We clarified this in the Results section L375 as:

> When comparing the aforementioned observed minimum soil moisture at a similar matric potential of the different treatments shows that biochar amendments had generally 7% (Period I), 4% (Period II) to 2 % (Period III) higher minimum soil water content (Figure 4). In Period II, only BC1 had a 4% minimum soil moisture at a similar matric potential while BC2 and C were similar.

**References**

Allen, S. T., Kirchner, J. W., Braun, S., Siegwolf, R. T. W. and Goldsmith, G. R.: Seasonal origins of soil water used by trees, Hydrol. Earth Syst. Sci., 23(2), 1199–1210, https://doi.org/10.5194/hess-23-1199-2019, 2019.

Amin, A., Zuecco, G., Geris, J., Schwendenmann, L., McDonnell, J. J., Borga, M. and Penna, D.: Depth distribution of soil water sourced by plants at the global scale: A new direct inference approach, Ecohydrology, 13(2), e2177, https://doi.org/10.1002/eco.2177, 2020.

Bachmann, D., Gockele, A., Ravenek, J. M., Roscher, C., Strecker, T., Weigelt, A. and Buchmann, N.: No Evidence of Complementary Water Use along a Plant Species Richness Gradient in Temperate Experimental Grasslands, PLOS ONE, 10(1), 1–14, https://doi.org/10.1371/journal.pone.0116367, 2015.

Berry, Z. C., Evaristo, J., Moore, G., Poca, M., Steppe, K., Verrot, L., Asbjornsen, H., Borma, L. S., Bretfeld, M., Hervé-Fernández, P., Seyfried, M., Schwendenmann, L., Sinacore, K., De Wispelaere, L. and McDonnell, J.: The two water worlds hypothesis: Addressing multiple working hypotheses and proposing a way forward, Ecohydrology, 11(3), e1843, https://doi.org/10.1002/eco.1843, 2018.

Beyer, M., Koeniger, P., Gaj, M., Hamutoko, J. T., Wanke, H. and Himmelsbach, T.: A deuterium-based labeling technique for the investigation of rooting depths, water uptake dynamics and unsaturated zone water transport in semiarid environments, Journal of Hydrology, 533, 627–643, https://doi.org/10.1016/j.jhydrol.2015.12.037, 2016.

Blanco-Canqui, H.: Biochar and Soil Physical Properties, Soil Science Society of America Journal, 81(4), 687, https://doi.org/10.2136/sssaj2017.01.0017, 2017.

Brooks, R. J., Barnard, H. R., Coulombe, R. and McDonnell, J. J.: Ecohydrologic separation of water between trees and streams in a Mediterranean climate, Nature Geoscience, 3(2), 100–104, https://doi.org/10.1038/ngeo722, 2010.

Fischer, B. M. C., Manzoni, S., Morillas, L., Garcia, M., Johnson, M. S. and Lyon, S. W.: Improving agricultural water use efficiency with biochar – A synthesis of biochar effects on water storage and fluxes across scales, Science of The Total Environment, https://doi.org/10.1016/j.scitotenv.2018.11.312, 2018.

Fischer, B. M. C., Aemisegger, F., Graf, P., Sodemann, H. and Seibert, J.: Assessing the Sampling Quality of a Low-Tech Low-Budget Volume-Based Rainfall Sampler for Stable Isotope Analysis, Frontiers in Earth Science, 7, 244, https://doi.org/10.3389/feart.2019.00244, 2019.

Jiménez-Rodríguez, C. D., Coenders-Gerrits, M., Wenninger, J., Gonzalez-Angarita, A. and Savenije, H.: Contribution of understory evaporation in a tropical wet forest during the dry season, Hydrol. Earth Syst. Sci., 24(4), 2179–2206, https://doi.org/10.5194/hess-24-2179-2020, 2020.

Jin, H., Fischer, B. M. C., Rojas-Conejo, J., Köppel, C. J., Johnson, M. S., Morillas, L., Lyon, S. W., Durán-Quesada, A. M., Suárez-Serrano, A., Manzoni, S. and Garcia, M.: Quantifying upland rice growth and water use efficiency after biochar application with drone-based hyperspectral and thermal imagery, Agriculture, Ecosystems and Environment, In review, in review.

Koeniger, P., Marshall, J. D., Link, T. and Mulch, A.: An inexpensive, fast, and reliable method for vacuum extraction of soil and plant water for stable isotope analyses by mass spectrometry, Rapid Communications in Mass Spectrometry, 25(20), 3041–3048, https://doi.org/10.1002/rcm.5198, 2011.

Layman, C. A., Araujo, M. S., Boucek, R., Hammerschlag-Peyer, C. M., Harrison, E., Jud, Z. R., Matich, P., Rosenblatt, A. E., Vaudo, J. J., Yeager, L. A., Post, D. M. and Bearhop, S.: Applying stable isotopes to examine food-web structure: an overview of analytical tools, Biological Reviews, 87(3), 545–562, https://doi.org/10.1111/j.1469-185X.2011.00208.x, 2012.

Mahindawansha, A., Orlowski, N., Kraft, P., Rothfuss, Y., Racela, H. and Breuer, L.: Quantification of plant water uptake by water stable isotopes in rice paddy systems, Plant and Soil, 429(1), 281–302, https://doi.org/10.1007/s11104-018-3693-7, 2018.

Marshall, J., Cuntz, M., Beyer, M., Dubbert, M. and Kuehnhammer, K.: Borehole Equilibration: Testing a New Method to Monitor the Isotopic Composition of Tree Xylem Water in situ, Frontiers in Plant Science, 11, https://doi.org/10.3389/fpls.2020.00358, 2020.

McDonnell, J. J.: The two water worlds hypothesis: ecohydrological separation of water between streams and trees?, WIREs Water, 1(4), 323–329, https://doi.org/10.1002/wat2.1027, 2014.

Muñoz-Villers, L. E., Geris, J., Alvarado-Barrientos, M. S., Holwerda, F. and Dawson, T.: Coffee and shade trees show complementary use of soil water in a traditional agroforestry ecosystem, Hydrology and Earth System Sciences, 24(4), 1649–1668, https://doi.org/10.5194/hess-24-1649-2020, 2020.

Penna, D., Hopp, L., Scandellari, F., Allen, S. T., Benettin, P., Beyer, M., Geris, J., Klaus, J., Marshall, J. D., Schwendenmann, L., Volkmann, T. H. M., von Freyberg, J., Amin, A., Ceperley, N., Engel, M., Frentress, J., Giambastiani, Y., McDonnell, J. J., Zuecco, G., Llorens, P., Siegwolf, R. T. W., Dawson, T. E. and Kirchner, J. W.: Ideas and perspectives: Tracing terrestrial ecosystem water fluxes using hydrogen and oxygen stable isotopes – challenges and opportunities from an interdisciplinary perspective, Biogeosciences, 15(21), 6399–6415, https://doi.org/10.5194/bg-15-6399-2018, 2018.

Penna, D., Geris, J., Hopp, L. and Scandellari, F.: Water sources for root water uptake: Using stable isotopes of hydrogen and oxygen as a research tool in agricultural and agroforestry systems, Agriculture, Ecosystems & Environment, 291, 106790, https://doi.org/10.1016/j.agee.2019.106790, 2020.

Prechsl, U. E., Gilgen, A. K., Kahmen, A. and Buchmann, N.: Reliability and quality of water isotope data collected with a low-budget rain collector, Rapid Communications in Mass Spectrometry, 28(8), 879–885, https://doi.org/10.1002/rcm.6852, 2014.

Rothfuss, Y. and Javaux, M.: Reviews and syntheses: Isotopic approaches to quantify root water uptake: a review and comparison of methods, Biogeosciences, 14(8), 2199–2224, https://doi.org/10.5194/bg-14-2199-2017, 2017.

Sprenger, M. and Allen, S. T.: What Ecohydrologic Separation Is and Where We Can Go With It, Water Resources Research, 56(7), e2020WR027238, https://doi.org/10.1029/2020WR027238, 2020.

Sprenger, M., Herbstritt, B. and Weiler, M.: Established methods and new opportunities for pore water stable isotope analysis, Hydrological Processes, 29(25), 5174–5192, https://doi.org/10.1002/hyp.10643, 2015.

Sprenger, M., Leistert, H., Gimbel, K. and Weiler, M.: Illuminating hydrological processes at the soil-vegetation-atmosphere interface with water stable isotopes, Reviews of Geophysics, 54(3), 674–704, https://doi.org/10.1002/2015RG000515, 2016.

van der Velde, Y., Heidbüchel, I., Lyon, S. W., Nyberg, L., Rodhe, A., Bishop, K. and Troch, P. A.: Consequences of mixing assumptions for time-variable travel time distributions: Mixing assumptions and time-variable travel time distributions, Hydrological Processes, 29(16), 3460–3474, https://doi.org/10.1002/hyp.10372, 2015.

Volkmann, T. H. M., Kühnhammer, K., Herbstritt, B., Gessler, A. and Weiler, M.: A method for in situ monitoring of the isotope composition of tree xylem water using laser spectroscopy, Plant, Cell & Environment, 39(9), 2055–2063, https://doi.org/10.1111/pce.12725, 2016.

---

## Author Comment (AC2) · 8 Jan 2021

**Hess-2020-404    Author reply on comment from Anonymous Referee #2:**

We thank Reviewer #2 for the constructive feedback. Please find below our responses to the individual comments and suggestions (reviewer #2 comments in blue font, with our response in black font).

*The main objective of this study was to investigate plant water sources.*

*First, I should say that after the work of Brooks et al. (2020), countless studies across regions (including tropical wet environments; see Goldsmith et al. 2012; Muñoz-Villers et al. 2018) and revisions have showed that plants use evaporatively fractionated soil water (Sprenger et al., 2016; Sprenger and Allen 2020). The general finding is that plant water is isotopically similar to bulk soil water but not to low suctionˇARˇ lysimeter water, implying that roots are generally located in less conductive (mobile) pores where water tends to travel more slowly and can reside for longer times.*

We agree with Reviewer 2 and recognize the important work of the Brooks et al. 2010 and others which were also cited in our manuscript. Seeing the on-going discussion around which water plants use (Berry et al., 2018; Sprenger and Allen, 2020) and as presented in our manuscript that the majority of the plant water studies focused on trees in natural catchments. Instead, as stated in L110-113: isotopes have been used to a lesser extent in agricultural systems than in natural systems to investigate plant water sources (Penna et al., 2020). There are successful studies done in coffee (Muñoz-Villers et al., 2020), maize, wheat (Stumpp et al., 2009) and rice cultures (Mahindawansha et al., 2018; Shen et al., 2015). This shows the need to continue and explore the potential of stable isotopes of the water in ecohydrological studies, especially in agricultural settings and novel aspect of this study.

As described in L103 the soil water can consist of different pools of water: "…plants use mobile vs. immobile soil water pools (Brooks et al., 2010)…". To infer which water plants use, lysimeters are commonly used to sample mobile soil water(Sprenger et al., 2015). Instead to access the tighter bound and more fractioned soil water with matric potential <-0.1 MPa, the immobile soil water, it is necessary to use e.g. the cryogenic vacuum method (Sprenger et al., 2015). Therefore, we agree with the reviewer that all soil water collected with lysimeters should be considered mobile soil water.

To highlight and avoid confusion we distinguish between mobile and immobile water in the new version of the manuscript.

*In the present study, soil water samples representing the soil water pool for plants, were only collected using low suction (80 kPa) lysimeters. Bulk soil samples were a key part of the experiment but they are missing here. This methodological issue, is perhaps the largest flaws of the research and I do not see a way to get out of it, based on all the evidence published over the last decade.*

*In addition, your soils are dominated by clay content (Table A1) which is very well known for its very fine particle structure making very difficult to empty the water from such smaller pores using low soil tension lysimeters. This is other reason why the authors should have collected and used bulk soil water isotope ratios (from cryogenic vacuum distillation) as the representative soil water source for plants.*

We agree with the reviewer that soil water collected with lysimeters should be considered mobile water (consistent with the comment from Reviewer #1). To have information on the full spectrum of the mobile and immobile soil water we did collect soil cores for bulk soil water extraction, especially in low

matric potential situations as in Period III. On average, however, we could extract less than 0.1 ml of water per soil sample using cryogenic vacuum extraction. This was too little water for pipetting and analysis in the LIS-autosampler setup. As a result, we could not gain information on the immobile water isotopic composition. As a result, we could not gain information on the bulk isotopic composition. Initially we decided to not included this information in the manuscript but feel to it is necessary include this in the method section of our manuscript and

After Old L224:

> In addition to the lysimeters, soil samples for bulk soil water extraction and subsequent stable isotope analysis in were collected randomly in all plots from a depth of ~10 cm on 7 out of 11 sampling days. In order to not disturb the rice plants, instead of an auger, a steel rod 50 cm in length with a 2cm diameter was pushed 10 cm into the soil. After removing 5 cm of the topsoil, the soil sample was collected (~5 cm ∅2cm). The sample was then placed in a double re-sealable zipper storage bag. To minimize post-sampling evaporation, the storage bags were directly placed in a cooler with ice. All soil samples were stored in the laboratory freezer (-80 °C) before extracting the soil water for isotopic analysis.

And old L235-246:

> Plant water was extracted from the stem (xylem water) of the different rice plants to infer which sources of water the rice plants used. We used the cryogenic vacuum extraction technique described by Koeniger et al., (2011) to extract the plant and bulk soil water for stable isotope analysis. The method uses a heated vial and a cold trap vial (Exetainer® vial with standard cap and rubber septum, Labco Ltd, Lampeter, United Kingdom) connected with stainless-steel capillary tubing. About 3 g of plant material from the rice stem was placed in the heated vial before the system was evacuated to 85 kPa with a vacuum hand pump (Mityvac). The heated vial was heated for 1 hour at 100°C using a test tube heater (HI839800 COD Test Tube Heater; Hanna instruments) while the cold trap vial rested in a Dewar flask containing liquid nitrogen at about -196°C. After the extraction was stopped, the cold trap vial was sealed with Parafilm and left to thaw. After thawing, the extracted liquid water was pipetted into 2 ml vials (32 x 11.6 mm screw neck vials with cap and PTFE/silicone/PTFE septa) and stored refrigerated (5 °C) until stable isotope analysis. The plant root and bulk soil water was extracted in the same manner as the xylem water using the cryogenic vacuum extraction technique but with extraction time longer than 3 hours. On average 86±5 % plant water and soil water were extracted. However, we extracted less than 0.1 ml of water per soil sample for the bulk soil water and less than 0.1 ml of water per root sample for the root water which were too small volumes for pipetting and the LIS-autosampler setup.

We felt due to the missing bulk soil water information to not calculate the fraction of different plant water sources and over interpret our data. However, to comply with the Reviewer #1 request we quantified, based on the available isotopic data, the fraction of different plant water sources to better compare the plant water use in biochar amended treatments with the control treatment. Mixing models are powerful tools to estimate the plant water sources (Layman et al., 2012; Rothfuss and Javaux, 2017).

The end-member mixing model will be presented in the method section after L328 as:

In addition, in the different treatments the potential plant water use of the rice plants was quantified using mixing model. Mixing models are powerful tools to estimate the plant water sources (Layman et al., 2012; Rothfuss and Javaux, 2017). However, applying Bayesian mixing models would not decrease the uncertainty in potential plant water sources due to the missing bulk soil water isotope data due to the little amount of cryogenically extracted bulk soil water. Instead, a simple three end-member mixing model was used:

$$O_{PW} = f_R O_R + f_{SI} O_{SI} + f_{SM} O_{SM} \qquad (9)$$

$$D_{PW} = f_R D_R + f_{SI} D_{SI} + f_{SM} D_{SM} \qquad (10)$$

$$1 = f_R + f_{SI} + f_{SM} \qquad (11)$$

$$f_R = \frac{O_{PW} D_{SM} - O_{PW} D_{SI} + O_{SM} D_{SI} - O_{SM} D_{PW} + O_{SI} D_{PW} - O_{SI} D_{SM}}{O_R D_{SM} - O_R D_{SI} + O_{SM} D_{SI} - O_{SM} D_R + O_{SI} O_R - O_{SI} D_{SM}} \qquad (12)$$

$$f_{SI} = \frac{O_{PW} D_{SI} - O_{PW} D_R + O_R D_{PW} - O_R D_{SI} + O_{SI} D_R - O_{SI} D_{PW}}{O_R D_{SM} - O_R D_{SI} + D_{SM} D_{SI} - D_{SM} O_R + O_{SI} D_R - O_{SI} D_{SM}} \qquad (13)$$

$$f_{SM} = \frac{O_{PW} D_R - O_{PW} D_{SM} + O_R D_{SM} - O_R D_{PW} + O_{SM} D_{PW} - O_{SM} D_R}{O_R D_{SM} - O_R D_{SI} + O_{SM} D_{SI} - O_{SM} D_R + O_{SI} D_R - O_{SI} D_{SM}} \qquad (14)$$

where $f_R$, $f_{SI}$, $f_{SM}$ [-] represents the fraction of plant water source of soil water from antecedent rainfall, immobile and mobile soil water while $O$ and $D$ indicate $\delta^{18}O$ and $\delta^2H$ [‰] respectively where subscript PW, R, SI and SM indicate the plant water, soil water with signature from antecedent rainwater, immobile and mobile soil water respectively. Seeing the limitation of the missing immobile water the three end-member mixing model was rather used as an explorative approach to support the more descriptive data analysis to explain potential plant water use of the rice plants in the different treatments.

[Figure]

*Figure 9   The fraction of plant water source rainfall (top), soil water from antecedent rainfall (middle) and the mobile soil water (bottom) indicated for each plant water sample (circle), the median fraction of the treatment (red line) and box plots (n>5) for the different treatments BC1, BC2 and C (indicated with grey letters) for the different sampling days 1-11. The boxes show the range of values for different sample groups (showing the median and the interquartile range, with whiskers indicating 10th and 90th percentiles). Italic grey numbers on the top panel indicate the number of times it was possible to calculate the fraction of different plant water sources out of the total of plant water samples of a treatment on a sampling day.*

[Figure]

*Figure A4 The stable isotope composition δ¹⁸O (left) and δ²H (right) for precipitation (P), immobile water (SI), mobile water (SM) and the plant water of the BC1, BC2 and control treatment for the sampling days 1-6 (rows). Circles indicate the data points. The boxes show the range of values for different sample groups (showing the median and the interquartile range, with whiskers indicating 10th and 90th percentiles). The red bar indicates the used end-member used in the mixing model (P-SM) and the median value for BC1, BC2 and C.*

[Figure]

*Figure A4 (continue) The stable isotope composition δ¹⁸O (left) and δ²H (right) for precipitation (P), immobile water (SI), mobile water (SM) and plant water of the BC1, BC2 and control treatment for the sampling days 7-11 (rows). Circles indicate the data points. The boxes show the range of values for different sample groups (showing the median and the interquartile range, with whiskers indicating 10th and 90th percentiles). The red bar indicates the used end-member used in the mixing model (P-SM) and the median value for BC1, BC2 and C.*

The results and limitation were discussed in 5.3 Temporally variable plant water sources after L510:

The rice plants in this study had different water sources available during different periods of the experiment, but what water did they consume?

Mixing model results in Period I (Figure 9), indicated that it is likely that the plants consumed dominantly immobile. This is consistent with results observed in previous studies using stable water isotopes to map out plant water sources (Brooks et al., 2010; Penna et al., 2020; Sprenger et al., 2016). This interpretation of plant water composition is supported by plant water samples falling along the theoretical evaporation lines estimating how soil water would evolve isotopically due to evaporation (Figure 7). Therefore, it is likely that during Period I, the young rice plants (with shallow root system <20 cm as reported by Mahindawansha et al. 2018) consumed the immobile (Figure 7 and 9) which was not sampled with the lysimeters at 15 cm

and 40 cm below the surface. We could, unfortunately, cannot confirm this as we could not extract enough bulk soil water for isotopic analysis.

During Period II, plants grew to their maximum heights with roots reaching deeper soil layers (depth >60 cm as reported by Mahindawansha et al. 2018). This means that the rice plants, similar to larger vegetation (Allen et al., 2019), would have had access to deeper and more-stable pools of water with a distinct lower *d-excess* signature. However, the isotopic composition of plant water during this period followed the GMWL (Figure 7 b, e and h) and mixing model results (Figure 9), indicating that plants consumed largely shallow soil water from recent rainfall.

In Period III, it became increasingly difficult to extract water from lysimeters at 15 cm below the surface and the isotopic composition of plant water drifted from the GMWL, falling along the theoretical evaporation line of residual rainfall falling in Period II (Figure 7 l, o and r) which is supported by the mixing model results (Figure 9). With the experiment being held in the tropics and based on the findings from Amin et al (2020) one would expect that the rice plants with their longer roots would accessed the more stable and older water stores in deeper subsurface zones below 60 cm. Instead, the rice plants in the different treatments preferably consumed the temporally variable and "newer" surface near soil water from recent rainfall similarly to what has been documented in grasslands (Bachmann et al., 2015) and catchments (e.g. van der Velde et al., 2015).

As previously stated, mixing models are only as good as the available data. Despite the draw back and source of uncertainty in the simple mixing model due to the missing bulk soil water isotopic composition, the results were still useful as explorative tool to support the more qualitative data analysis (Figure 5-8 and A4). Our results provided a first insight that plants water sources were largely variable within a treatment but no difference between biochar and control treatments could be observed. Seeing the large uncertainty and variability of plant water sources demonstrates the need of further and more detailed research of plant water use in biochar amendments. By performing more detailed isotopic experiments (Beyer et al., 2016), higher temporal resolution sampling of plant water (Marshall et al., 2020; Volkmann et al., 2016) and spatiotemporal soil water (Sprenger et al., 2015) or including interception, transpiration and atmospheric processes into the experimental analysis (Jiménez-Rodríguez et al., 2020) which would allow to not only distinguish whether the rice plants prefer mobile or . immobile water (Berry et al., 2018; Brooks et al., 2010; McDonnell, 2014; Muñoz-Villers et al., 2020) but also to more accurately quantify the fraction of water sources. Next to the aforementioned vertical processes, the lateral water fluxes (Sprenger and Allen, 2020) need to be considered to assess the field-scale responses to biochar amendments (Fischer et al., 2018) which would allow to better constrain the dominant ecohydrological process as e.g. Muñoz-Villers et al.(2020). Furthermore, when mixing biochar in the top soil, a multi-layer soil profile is created and based on studies in natural catchments, e.g. Penna et al. (2018) or Sprenger et al. (2016), these different layers could store not only different quantities of water but also water characterized by different ages. In addition, Blanco-Canqui (2017) discusses how biochar can age thereby altering the physical and chemical characteristics of biochar and soil in time. The long-term effect of biochar soil amendments in tropical agroecosystems would also need

to be considered especially to understand and identify which sources of water the rice plants consume from year to year.

Although these more detailed short and long-term analyses were beyond the scope of this initial investigation our results from one growing season indicate that rice plants growing in biochar amended soils not only had access to more water (Figure 4) but also had a more stable source of green water (i.e., soil moisture from rainfall) and thus could withstand dry spells seven days longer (Figure 3). Regardless of the potential advantages, as stated by Fischer et al. (2018), it must be noted that biochar as water management tool does not adhere to a one size fits all approach but needs fine tuning in accordance with climate, site and plant characteristics to obtain stable and optimal yields.

*I also observed that your samplings 9,10 and 11 corresponding to Period III (Figure 4), were characterized by low soil water contents held at very high tensions (close to PWP conditions). Hence, the soil water collected with soil lysimeters was not "seeing" the water that plants were extracted during this dry period. This situation is particularly observed for the biochar amended treatments. Therefore, the research question 2 cannot be answered.*

It is correct that not all lysimeter could extract water during the dry period III. This off course does not imply that plants cannot extract water from the soil matrix as also demonstrated by different other studies e.g. (Brooks et al., 2010). We agree also with the reviewer that soil water collected with lysimeters is considered mobile water and as stated previously we will highlight this term in the manuscript.

Without overinterpreting the outcome of the mixing model, as stated, we see the use of the mixing model rather as supportive and explorative tool.

The results of the qualitative data analysis and the simple mixing model show similar results:

1)  The isotopic composition of plant water has a large variability which is similar in all three treatments (Figure 5 and 6).
2)  The isotopic composition of plant water has a large variability in the dual isotope space and similar temporal pattern in the different treatments (Figure 7) which can be related plant water use from soil moisture from resent rainfall, fractionated soil moisture from recent rainfall and mobile soil water.
3)  The results of the mixing model show quantify the proportions of the aforementioned different plant water sources and show similarly a large variability.

Despite the draw back and source of uncertainty in the simple mixing model, caused by the missing bulk soil water isotopic composition, when used with care the results were a useful as explorative tool to support the more qualitative data analysis (Figure 5-8 and A4). Overall, it shows that the isotopic composition of the plant water samples across different treatments had a large within treatment variability but small difference between treatments. Which answers our research question 2) do rice plants grown in biochar amended soils access different pools of water compared to those grown in non-amended soils?

However, we also highlight that seeing the large uncertainty and variability of plant water sources shows the need of further and more detailed research of plant water use in biochar amendments.

*I have made some other important comments that the authors can also consider when preparing other articles around these topics:*

*1) The use of mixing models to quantify the relative contributions of the different plant water sources, instead of reporting the results in a visual graphical and/or descriptive way only.*

Similar to Reviewer #1, we agree and thank for Reviewer #2 pointing us that we presented and described our collected data rather qualitatively. We felt that due to the lack of bulk soil water information it would be rather uncertain and potentially erroneous to calculate the fraction of different plant water sources, which could lead us to over interpret our data. However, we agree that a better job needs to be done quantifying potential plant sources to avoid being overly qualitative. Thus, we have quantified, based on the available isotopic data, the fraction of different plant water sources to better compare the plant water use in biochar amended treatments with the control treatment.

Although we see the potential of Bayesian mixing models we agree with Layman et al. (2012) statement that "including Bayesian-based approaches, are not a quick fix or a substitute for poor sampling strategy". Or as in our case, applying Bayesian mixing models would not decrease the uncertainty in potential plant water sources due to the missing bulk soil water isotope data which was caused due to the little amount of cryogenically extracted bulk soil water. However, from the descriptive analysis of the collected isotope data could identify different end-members. We identified that soil water from recent rainfall as a first potential plant water source (Figure 5-8 and A4). Seeing the rather small range (<1 ‰ $\delta^{18}O$ and <5‰$\delta^{2}H$) and partly overlapping isotopic composition of mobile soil water and ground water (Figure 5, 6, 8 and A4), we chose the median isotopic composition of soil water collected at -15 cm as a potential second plant water source. In addition, we observe in Figure 7 that the plant water samples drifted in time from the GMWL along the theoretical evaporation line of the soil water from antecedent rainfall. From this, we assume that the end point of the theoretical evaporation line can serve as proxy of the third potential water source if plants access tighter bound soil water -the immobile water. These three end-members were used in a simple three-component mixing model to explore and support which sources of water the rice plants potentially consumed during the different sampling days.

*2) The construction of dual isotope space figures in which the plant and the different water sources are plotted together. In this way, it is easier the assess the isotope information per sampling period and seasons (Figure 7 and 8).*

We agree and now included next to the isotopic composition of single rainfall samples also for each sampling day the mobile soil water and ground water data in the new Figure 7.

[Figure]

*Figure 7   The dual isotope space with the isotopic composition of daily rainfall samples (crosses), plant water samples (circles), the calculated evaporation lines of residual rainfall and sampled soil water for the treatments BC1 (a-c), BC2 (d-f) and C (g-i) and periods I-III (columns). Colors indicate the different sampling days (note that lines in period III are blue because they have been obtained from samples taken in period II). The local meteoric line (black dotted line) and global meteoric water line (grey solid line) are indicated in all panels. The grey dashed lines (panel a, d and g) indicate the evaporation line of median soil water. Isotopic compositions of irrigation, soil water and groundwater vary within the grey shaded squares indicated as 8 j-8 r, and enlarged in figure 8 j-r.*

*3) Both water isotopes (_2H and _18O) were determined for the plant and potential water sources, however, the results were only elaborated around 18O. I would suggest you to describe both isotope ratios.*

We included now also $\delta^2H$ in the text.

[revised manuscript text omitted]